METHODS AND RESOURCES

# Cell-based and multi-omics profiling reveals dynamic metabolic repurposing of mitochondria to drive developmental progression of *Trypanosoma brucei*

Eva Doleželová[1], Michaela Kunzová[1,2], Mario Dejung[3]☯, Michal Levin[3]☯,
Brian Panicucci[1], Clément Regnault[4], Christian J. Janzen[5], Michael P. Barrett[4],
Falk Butter[3], Alena Zíková[1,2]*

**1** Institute of Parasitology, Biology Centre, Czech Academy of Sciences, Ceske Budejovice, Czech Republic,
**2** Faculty of Science, University of South Bohemia, Ceske Budejovice, Czech Republic, **3** Institute of
Molecular Biology (IMB), Mainz, Germany, **4** Welcome Centre for Integrative Parasitology, Institute of
Infection, Immunity and Inflammation, Glasgow Polyomics, College of Medical, Veterinary and Life Sciences,
University of Glasgow, Glasgow, United Kingdom, **5** Department of Cell and Developmental Biology,
Biocenter, University Wuerzburg, Wuerzburg, Germany

☯ These authors contributed equally to this work.
* azikova@paru.cas.cz

Bern, SWITZERLAND

**Data Availability Statement:** All relevant data are
within the paper and in S1 Data file. Transcriptomic
data are available at https://www.ncbi.nlm.nih.gov/

## Abstract

Mitochondrial metabolic remodeling is a hallmark of the *Trypanosoma brucei* digenetic life
cycle because the insect stage utilizes a cost-effective oxidative phosphorylation (OxPhos)
to generate ATP, while bloodstream cells switch to aerobic glycolysis. Due to difficulties in
acquiring enough parasites from the tsetse fly vector, the dynamics of the parasite's meta-
bolic rewiring in the vector have remained obscure. Here, we took advantage of in vitro–
induced differentiation to follow changes at the RNA, protein, and metabolite levels. This
multi-omics and cell-based profiling showed an immediate redirection of electron flow from
the cytochrome-mediated pathway to an alternative oxidase (AOX), an increase in proline
consumption, elevated activity of complex II, and certain tricarboxylic acid (TCA) cycle
enzymes, which led to mitochondrial membrane hyperpolarization and increased reactive
oxygen species (ROS) levels. Interestingly, these ROS molecules appear to act as signaling
molecules driving developmental progression because ectopic expression of catalase, a
ROS scavenger, halted the in vitro–induced differentiation. Our results provide insights into
the mechanisms of the parasite's mitochondrial rewiring and reinforce the emerging concept
that mitochondria act as signaling organelles through release of ROS to drive cellular
differentiation.

## Introduction

In most eukaryotic cells, energy metabolism is an interplay between two energy production
pathways to generate ATP: an efficient mitochondrial oxidative phosphorylation (OxPhos)

geo/query/acc.cgi?acc=GSE140796. Proteomic data are available at the ProteomeXchange Consortium via the PRIDE, with the dataset identifier PXD016370. Metabolomic data are available at MetaboLights database with identifier MTBLS1390.

**Funding:** This work was supported by grants from the Grant Agency of the Czech Republic (17-22248S) and by the ERD fund (No. CZ.02.1.01/0.0/0.0/16_019/0000759), both to AZ, as well as by Grant Agency of the University of South Bohemia (GAJU 080/2020/P) to MK. MPB is funded as part of the Wellcome Trust core grant to the Wellcome Trust Centre for Integrative Parasitology (WCIP) - 104111/Z/14/Z. The funders had no role in study design, data collection and analysis, decision to publish, or preparation of the manuscript.

**Competing interests:** The authors have declared that no competing interests exist.

**Abbreviations:** AAC, ATP/ADP carrier; ACL, ATP-dependent citrate lyase; AOX, alternative oxidase; BARP, brucei alanine-rich protein; BNE, blue-native electrophoresis; BPS, bathophenanthroline disulphonic acid; BSF, bloodstream form; CS, citrate synthase; DCF, 2′,7′-dichlorofluorescein; DDA, data-dependent acquisition; ETC, electron transport chain; EP, procyclin rich in Glu-Pro repeats; FAD, flavin adenine dinucleotide; FBPase, fructose bisphosphatase; FBS, fetal bovine serum; FCCP, carbonyl cyanide-4-phenylhydrazone; GLM, generalized linear model; GO, Gene Ontology; GPEET, procyclin rich in Gly-Pro-Glu-Glu-Thr repeats; hrCNE, high-resolution clear-native electrophoresis; hsp70, heat shock protein 70; H$_2$DCFHDA, 2′,7′-dichlorofluorescein diacetate; IDH, isocitrate dehydrogenase; IF1, inhibitory peptide 1; k, kinetoplast; KCN, potassium cyanide; kDNA, kinetoplast DNA; LC-MS, liquid chromatography–mass spectrometry; LFQ, label-free quantification; mAb, monoclonal antibody; MCU, mitochondrial calcium uniporter; mVSG, metacyclic-like variable surface glycoprotein; n, nucleus; NGS, next generation sequencing; OxPhos, oxidative phosphorylation; p, posterior end; pAb, polyclonal antibody; PBS, phosphate-buffered saline; PCA, principal component analysis; PCF, procyclic form; PiC, phosphate carrier; RBP6, RNA binding protein 6; RBP6$^{OE}$, RBP6 overexpressing; RET, reverse electron transfer; ROS, reactive oxygen species; SBPase, sedoheptulose-1,7-bisphosphatase; SCoAS, succinyl Co-A synthetase; SDS, sodium dodecyl sulfate; SHAM, salicylhydroxamic acid; SOD, superoxide dismutase; TCA, tricarboxylic acid; THT, *Trypanosoma brucei* hexose transporter; TMRE, tetramethyl rhodamine ethyl ester.

and an ancient glycolytic pathway. A "textbook-like" aerobic eukaryotic cell relies mainly on cost-effective OxPhos to fulfill cellular requirements for ATP. Exceptions apply to some rapidly proliferating cells, e.g., some cancer cells exploit high rates of fermentative glycolysis, irrespective of oxygen availability [1]. Oncogenic metabolic reprogramming from OxPhos to glycolysis allows the malignant cell to fulfill a great demand for synthesis of nucleotides and amino acids, the building blocks of DNA and proteins, respectively, to promote tumor growth and progression [2]. Metabolic rewiring to aerobic glycolysis also drives the activation of macrophages to a pro-inflammatory phenotype in response to infection. This phenotypic switch involves suppressed OxPhos, disruption of the tricarboxylic acid (TCA) cycle, reactive oxygen species (ROS) production, and accumulation of succinate [3]. The factors that underlie the striking metabolic changes during the aforementioned cellular processes are not fully understood. However, multiple lines of evidence implicate alterations in mitochondrial function that lead to the release of signal molecules to drive cell differentiation. Prominent examples of these signals are ROS and certain metabolic intermediates (e.g., succinate, citrate, and itaconate) with the ability to affect gene expression at the global level via posttranslational mechanisms [4–6]. This demonstrates how mitochondrial plasticity and metabolic remodeling are crucial for cells to respond to various signals to acquire new functions.

A quintessential example of metabolic remodeling is represented by metabolic changes underlying the life cycle of a digenetic mammalian parasite, *Trypanosoma brucei* [7]. This parasite of medical and veterinary importance needs to make crucial adaptations to new environments, including different temperatures and nutrients, as it alternates between an insect vector, the tsetse fly, and a mammalian host [8]. Dictated by the availability of nutrients in the midgut of the tsetse fly, the insect procyclic form (PCF) accumulates and metabolizes amino acids, including proline and threonine, using an incomplete TCA cycle, which leads to the production of different metabolic intermediates as well as ATP by substrate-level phosphorylation [9,10]. The cells respire through a canonical cytochrome-containing electron transport chain (ETC), which generates a mitochondrial membrane potential (Δψm) that powers the F$_o$F$_1$-ATP synthase [11,12]. The full activity of the parasite's mitochondrion is reflected by its highly branched, reticulated structure that intertwines the whole cell. Bloodstream form (BSF), by contrast, inhabits glucose-rich blood of the mammalian host, and the majority of its ATP is produced by a highly active glycolytic pathway, the redox balance of which is maintained by a glycosomal glycerol-3-phosphate/dihydroxyacetone phosphate shuttle via mitochondrial flavin adenine dinucleotide (FAD)-dependent glycerol-3-phosphate dehydrogenase and alternative oxidase (AOX) [13,14]. In the absence of ETC complexes III and IV, the Δψm is generated by ATP hydrolysis through the F$_o$F$_1$-ATP synthase complex operating in its reverse mode [15,16]. Structurally, the BSF mitochondrion is a simple tubular organelle lacking recognizable invaginations. Over the past 25 years, we have acquired extensive knowledge of PCF and BSF metabolism due to the availability of axenic cultures [17,18]. However, the molecular mechanisms underlying the dramatic metabolic remodeling that occurs during differentiation, including signals and signal transduction pathways, are elusive. This lack of knowledge is mainly due to difficulties in characterizing parasites undergoing differentiation from one life cycle stage to another in the tsetse fly [19].

Upon entry into a tsetse fly during a blood meal, the tsetse-primed (stumpy form) subpopulation of the BSF trypanosomes responds to two major signals: lower temperature and the presence of metabolites (e.g., *cis*-aconitate, citrate) to trigger differentiation to the PCF morphotype [20]. Once the PCF cells establish infection in the tsetse midgut, the parasites migrate to the ectoperitrophic space by crossing the acellular peritrophic membrane and re-enter the alimentary canal at the proventriculus as elongated trypomastigotes [21]. These cells differentiate to short epimastigotes through an asymmetric division that produces one long and one

short daughter. The short epimastigotes are believed to colonize salivary glands, where they fully differentiate to attached epimastigotes. Early during infection, these attached morphotypes divide symmetrically into identical progeny and colonize salivary glands. Later, the epimastigote cells undergo asymmetric division, producing a daughter cell of the metacyclic type. These cells are released to the lumen and are ready to be transmitted during the next blood meal to establish infection in a new mammalian host [22–24]. Until recently, the tsetse morphotypes were not available in culture, restricting our knowledge about their metabolism and signaling pathways underlying their complex developmental program.

The discovery that overexpression of a single RNA binding protein 6 (RBP6) induced differentiation of PCF cells to epimastigotes and further to infective metacyclics [25,26] offered a simplified route to dissect differentiation-related events. Using this system, we report here a multi-omics and cell-based profiling to describe the metabolic alterations that accompany in vitro–induced differentiation. We provide a series of publicly available multi-omics datasets that will be of high relevance to the community interested in exciting aspects of *Trypanosoma* biology and biological data integration. Moreover, our data show dynamic metabolic remodeling of mitochondria from cytochrome-mediated respiration to AOX, followed by increased Δψm and ROS production to drive the developmental progression of *T. brucei* tsetse life cycle stages.

## Results

### Establishment of RBP6^OE cell line

For the study reported here, we took advantage of an in vitro differentiation system based on the inducible expression of RBP6 in the *T. brucei* Lister 427 (29–13) strain [25]. As with published data, we observed a time-dependent appearance of epimastigotes, and then metacyclic cells (Fig 1A) in RBP6 overexpressing (RBP6^OE) trypanosomes adapted to no-glucose medium SDM-80 supplemented with N-acetyl glucosamine. The life cycle stages were determined by means of the stage-specific morphology based on shape and cell size, as well as the relative position of the kinetoplast to the nucleus. While the mitochondrial DNA called kinetoplast (kDNA) in PCF is posterior to the nucleus, during epimastigote maturation, it migrates to the opposite side of the nucleus, where it is found in its close proximity or at the anterior part of the cell. The metacyclic trypomastigotes are typically smaller than PCF and epimastigotes, and are characterized by the kinetoplast occupying the very end of the rounded posterior tip (Fig 1A). Furthermore, there are well-characterized differences at the level of the parasite surface molecules. The procyclic trypomastigotes are covered with a glycoprotein coat composed mainly of procyclins rich in Gly-Pro-Glu-Glu-Thr repeats (GPEET procyclin) and procyclins rich in Glu-Pro repeats (EP procyclin) [27], whereas mature epimastigote forms have a coat consisting of glycosylphosphatidyl inositol-anchored proteins, brucei alanine-rich proteins (BARPs) [28].

After the RBP6 overexpression (RBP6^OE), a cell was considered an epimastigote form if the kDNA was found juxtanuclear or at the anterior end of the nucleus. We also measured the distance between the nucleus and kDNA for each time point and detected a significant shift toward the nucleus at day 2 upon RBP6^OE (Fig 1B). Immunofluorescence analysis showed that at day 2, procyclic as well as epimastigote cells express procyclin on their surface (Fig 1C), while later during development, RBP6^OE cells expressed BARP (Fig 1D). A metacyclic cell was defined by a smaller size, having the kinetoplast in close proximity to the plasma membrane, and by uptake of fluorescein-labeled dextran due to metacyclic cells' up-regulated endocytosis [25] (Fig 1A). On day two after the RBP6 induction, the cell culture contained mainly epimastigotes (procyclin positive), and metacyclics were first detected on day 4. By day six, the culture

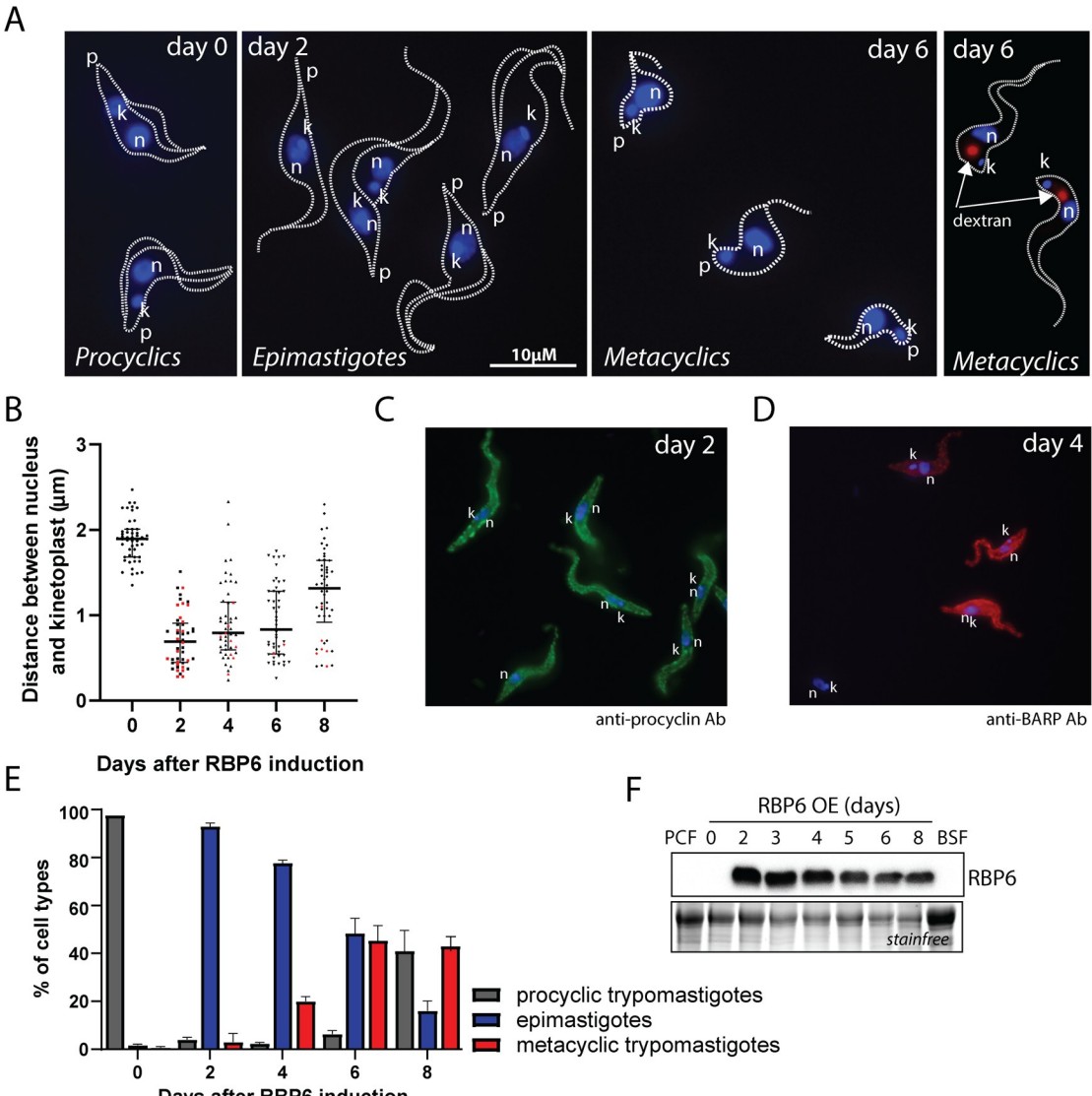

**Fig 1. RBP6-induced differentiation of PCF *T. brucei* cells in the absence of glucose.** (A) Imaging of procyclic, epimastigote, and metacyclic forms. Cells were characterized by means of morphological features such as shape, size, and the relative position of the kinetoplast to the nucleus, as well as by internalization of fluorescently labeled dextran. DNA was visualized by DAPI staining. (B) Quantification of the intracelullar distance between kDNA and nucleus. For each day, 50 cells were used for measurements. The red dots indicate cells with anterior localization of their kDNA. (C) Immunofluorescence analysis of RBP6$^{OE}$ cells at day 2 using anti-procyclin antibody. (D) Immunofluorescence analysis of RBP6$^{OE}$ cells at day 4 using anti-BARP antibody. (E) Time line for the appearance of epimastigotes and metacyclic cells upon induction of RBP6$^{OE}$. (F) Western blot analysis of whole-cell lysates from RBP6$^{OE}$ cells. Underlying data plotted in panels B and E are provided in S1 Data. Ab, antibody; BARP, brucei alanine-rich protein; BSF, bloodstream form; k, kinetoplast (mitochondrial DNA); kDNA, kinetoplast DNA; n, nucleus; p, posterior end; PCF, procyclic form; RBP6, RNA binding protein 6.

contained epimastigotes (mainly BARP positive) and the metacyclic form constituted approximately 50% of the total number of parasites (Fig 1E). On day 8, the culture comprised a mixture of nondividing metacyclics and epimastigotes, which were being overgrown by an emerging population of procyclin-positive procyclic-like cells. These cells may represent de-differentiating cells or else a population of procyclic cells that did not respond to RBP6 overexpression. RBP6 overexpression was verified by western blot (Fig 1F).

## Time-course transcriptomes and proteomes of RBP6[OE] trypanosomes

To determine global changes to the transcriptome and proteome across the in vitro–induced developmental progression of RBP6[OE] cells, we applied RNA-Seq and label-free quantitative proteomic analyses (Fig 2A).

RNA was extracted from cells induced for 0, 2, 3, 4, 6, and 8 days and processed using polyA-enrichment for sequencing on an Illumina NextSeq 500 platform. Our analysis of the transcriptomes is based on four biological replicates for each time point. The principal component analysis (PCA) showed a clear difference between the uninduced cells at day 0 and the later stages (S1 Fig). Furthermore, the transcriptomic analysis revealed significant (corrected *P* value <0.05) differential expression for 3,439 transcripts/genes during the RBP6-induced development (S1 Table). For example, at day 2 post-RBP6[OE], 1,221 transcripts were up-regulated and 1,588 transcripts were down-regulated when compared to the uninduced cells (Fig 2A), indicating a global change in transcript expression. First, we analyzed the expression profile of well-known surface molecule markers for early and late PCF (GPEET and EP procyclins, respectively) [29], mature epimastigotes (BARPs) [27], and infectious metacyclics, which express metacyclic-like variable surface glycoproteins (mVSGs) [30]. The GPEET transcript was highly down-regulated, while the procyclin EP transcripts and BARP transcripts were up-regulated immediately at day 2 following RBP6 induction. As expected, high levels of metacyclic specific mVSG transcripts were detected later during the differentiation process, namely at days 6 and 8 (Fig 2B). The presence of mRNAs for the three types of surface molecules at days 6 and 8 after the RBP6[OE] confirms the mixture of various life cycle stages (Fig 1E).

To identify groups of genes that followed similar trends, we performed time-course expression profiling based on K-medoids clustering and selected four clusters based on optimum average silhouette within the distance matrix (Fig 2C, S2 Table). Cluster 1 grouped genes being slowly down-regulated after RBP6 induction. Gene Ontology (GO) term analysis of this cluster indicated enrichment for genes involved in translation. This observation reflects the fact that metacyclic trypanosomes are metabolically quiescent, arrested in G1/G0 with repressed mRNA translation [26]. Cluster 2 is characteristic of genes being steadily up-regulated during developmental progression. Interestingly, GO annotation highlighted genes involved in signal transduction, regulation, and protein modification. Cluster 3 comprises genes being immediately down-regulated over the time-course experiments and contains genes involved in the regulation of gene expression, biosynthetic processes, and RNA processing. Cluster 4 shows fast up-regulation upon RBP6 induction, and it includes genes involved in oxidoreduction metabolic processes linked to the mitochondrion, such as transcripts for TCA cycle or proline degradation enzymes (Fig 2D, S2 Table).

Because transcriptomic data are available for several time-points of RBP6[OE] cells grown in the presence of glucose as well as for monomorphic populations of in vitro–differentiated metacyclics [26,31], we compared our time-course data with these previously published data-sets to benchmark developmental progression in our experiments. The RBP6[OE] time-course transcriptomes have high correlation coefficients (0.99) (S2 Fig, S3 Table) and become increasingly similar to metacyclic samples (S3 Fig).

For the proteomic analysis, we performed label-free quantitative mass spectrometry for the same time points as for the transcriptomes. Each time point consists of quadruplicate samples, measured with a 4-hour liquid chromatography (LC) gradient on a high-resolution mass spectrometer, and data were analyzed by MaxLFQ [32] (Fig 2A). We quantified 5,227 protein groups with a minimum of 2 peptides (1 unique) and present in at least two out of four replicates, covering differences in expression levels of three orders of magnitude (S4 Fig, S4 Table). PCA showed that replicates of the same time points clustered closely together, demonstrating

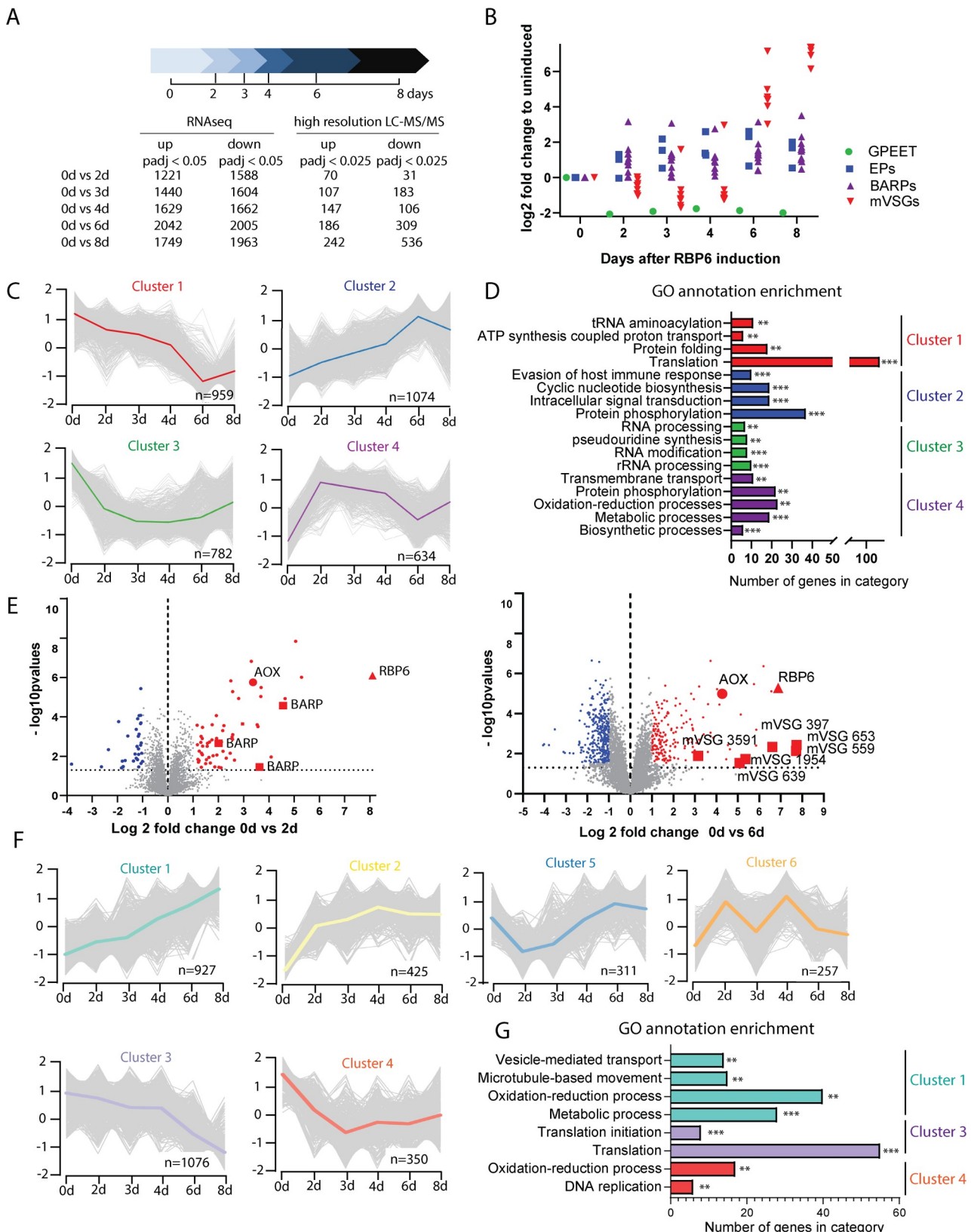

**Fig 2. RBP6^OE differentiation transcriptome and proteome.** (A) Scheme of the differentiation process and the experimental time points. The RBP6-induced cells were harvested and analyzed by RNA-Seq and label-free quantitative mass spectrometry at different time points, as indicated. The

table contains an overview of the number of differentially expressed transcripts and proteins. (B) Gene expression profile of transcripts encoding surface glycoproteins GPEET (Tb927.6.510), BARP (Tb927.5.15530, Tb927.5.15550, Tb927.5.15560, Tb927.5.15600, Tb927.5.15590), and *T. brucei* 427 mVSG (Tb427_000106600.1, Tb427_000524500.1, Tb427_000173600.1, Tb427_000288600.1, Tb427_000304300.1, Tb427_000108300.1, Tb427_000627000.1, Tb427_000615900.1). (C) Time-course expression profiling of transcriptomic data based on K-medoids clustering. (D) Gene Ontology (GO) term analysis applied to all four transcriptomics clusters. Significant levels are depicted as follows $^{**}P < 0.01$, $^{***}P < 0.0001$. (E) Volcano plots showing a comparison of protein expression levels (5,227 protein groups) between day 0 and day 2 (left panel) or day 6 (right panel) upon RBP6 induction. Log2 fold change values of averaged LFQ intensities from quadruplicate experiments are plotted against the respective −log10-transformed *P* values. Significantly down-regulated proteins are depicted in blue, while significantly up-regulated proteins are depicted in red. RBP6, AOX, and surface glycoproteins BARP (day 2) and metacyclic (m)VSGs (day 6) are highlighted. (F) Time-course expression profiling of proteomic data based on K-medoids clustering. (G) GO term analysis of Clusters 1, 3, and 4 showing a significant enrichment. $^{**}P < 0.01$, $^{***}P < 0.001$. Underlying data plotted in panel B are provided in S1 Data. AOX, alternative oxidase; BARP, brucei alanine-rich protein; EP, procyclin rich in Glu-Pro repeats; GO, Gene Ontology; GPEET, procyclin rich in Gly-Pro-Glu-Glu-Thr repeats; k, kinetoplast; LC-MS/MS, liquid chromatography tandem mass spectrometry; LFQ, label-free quantification; mVSG, metacyclic-like variable surface glycoprotein; n, nucleus; RBP6, RNA binding protein 6.

minimal experimental variation (S5 Fig). In fact, the separation of the time course is more clearly discernable at the proteome level (S1 Fig) than the transcriptome level (S1 Fig), indicating that posttranscriptional processes prevail in the progression towards metacyclics. The changes at the proteome level were relatively slow in emerging, because at day 2 only 31 proteins were significantly down-regulated (log2 fold change <−1, *P* value <0.025), while 70 proteins were significantly up-regulated (log2 fold change >1, *P* value <0.025). Among these proteins, RBP6 was detected together with BARPs and AOX (Fig 2E left panel, S4 Table). At day six, 495 proteins were differentially expressed, with mVSG glycoproteins being the most up-regulated entities detected (Fig 2E right panel, S4 Table).

To assess proteome remodeling systematically, we performed K-medoids clustering of the expression profiles across time and obtained six different clusters (Fig 2F), which were further analyzed for GO annotation enrichments (Fig 2G, S5 Table). Cluster 1 includes proteins that are being up-regulated during the developmental progression and, in agreement with the transcriptomic cluster 4, which exhibits a similar trend, it contains enzymes involved in energy metabolism in addition to proteins responsible for microtubule-based movement and vesicle-mediated transport. Cluster 3 is characterized by proteins being steadily down-regulated and contains mainly proteins involved in translation and translation initiation. Cluster 4 is enriched for proteins involved in DNA replication, DNA binding, and oxidation-reduction processes (S5 Table). Unfortunately, GO annotation analysis of clusters 2, 5, and 6 did not reveal any statistically significant enrichment. Cluster 2 comprises of proteins, which are being steadily up-regulated. Clusters 5 and 6 contain proteins with fluctuating expression (Fig 2F, S5 Table).

## Proteomics suggest mitochondrial metabolic alterations in proline oxidation pathway and TCA cycle over the *T. brucei* differentiation pathway towards metacyclogenesis

The remodeling of the energy metabolic pathways is a hallmark of *T. brucei* life cycle development. In the absence of glucose, PCF parasites metabolize proline in fully competent mitochondria, gluconeogenesis feeds the pentose phosphate pathway, and ATP is provided by OxPhos. In contrast, BSF cells generate the majority of cellular ATP by glycolysis, have a rudimentary mitochondrion, TCA cycle enzyme activities are barely detectable, and functional cytochrome *c*–mediated ETC is absent. An alternative truncated ETC involving glycerol-3-phosphate dehydrogenase and AOX sustains glycosomal redox balance, with oxygen being the electron sink [7,14].

To gain insight into the proposed rewiring from OxPhos to aerobic glycolysis, we checked the expression profile of glycolytic enzymes, subunits of respiratory chain complexes I, II, III,

and IV, the $F_oF_1$-ATP synthase, and TCA cycle enzymes over the induced differentiation pathway (Fig 3, S6 Fig). Some of the complex I subunits displayed only a slight increase, while the abundance of the $F_oF_1$-ATP synthase (complex V) subunits was largely unaffected (S6 Fig). Complex III and IV subunits showed slow but progressive down-regulation (S6 Fig). This detected trend would suggest that during the developmental progression, the cells maintain PCF-like mitochondrial metabolism, with the exception of the BSF final oxidase AOX, the expression of which was strongly elevated (Fig 3A) as cells progressed toward the bloodstream infective metacyclic form. Interestingly, expression of all 15 subunits of complex II was strongly up-regulated immediately at day two after RBP6 induction (Fig 3A). Because complex II is part of the TCA cycle, we examined other TCA cycle enzymes. Citrate synthase (CS), aconitase, and mitochondrial isocitrate dehydrogenase (IDH) also showed strong up-regulation (Fig 3A). Both proline transporters and mitochondrial dehydrogenases involved in proline catabolism were up-regulated, suggesting an unexpected spike in proline consumption during differentiation (Fig 3A). Interestingly, expression of the BSF-specific high-affinity proline transporter (Tb927.8.7610) was steadily up-regulated during differentiation. The activity of mitochondrial dehydrogenases is dependent on $Ca^{2+}$ ions, which are imported into the mitochondrion via a heterooligomeric mitochondrial calcium uniporter (MCU) [33,34]. Changes in expression in three out of four known MCU subunits were among the most notable features in our proteomic data, showing 6-fold up-regulation at day 6 following RBP6 induction (Fig 3A). Because the cells were grown in the absence of glucose, we assessed the expression profiles of gluconeogenetic enzymes. Interestingly, the majority of these were down-regulated or not affected except for sedoheptulose-1,7-bisphosphatase (SBPase) and fructose bisphosphatase (FBPase), the expression of which was up-regulated. Other enzymes involved in glucose metabolism were only mildly affected, including members of the pentose phosphate pathway. Noteworthy, the expression of *T. brucei* hexose transporters THT1 and THT2 were differentially altered. The PCF transporter, THT2, with low capacity and high affinity, showed a moderate down-regulation, while the high-capacity BSF transporter, THT1, was up-regulated later during the differentiation process, reaching >6-fold change at day 8 following RBP6 induction (Fig 3A), consistent with preadaptation for re-entry to the glucose-rich environment of the mammalian bloodstream. To validate our MS proteomic results, we performed western blot analysis for selected mitochondrial proteins with available antibodies and also included whole-cell lysate from in vitro cultured BSF cells (Fig 3B). As expected, we detected no significant changes in protein levels of the $F_1$-ATPase subunit beta, the ATP/ADP carrier (AAC), the phosphate carrier (PiC), or mitochondrial heat shock protein 70 (mitochondrial hsp70). We observed lower abundance of subunits of complexes IV and III, trCOIV, and Rieske. As measured by proteomics, aconitase, pyruvate dehydrogenase, subunit I of complex II (SDH66), succinyl-CoA synthetase (SCoAS; subunit beta), and AOX were up-regulated over the time course of developmental progression. To summarize the acquired data, Fig 4 highlights changes in enzymatic pathways during the in vitro–induced metacyclogenesis and proposes that the RBP6$^{OE}$ cells maintain their PCF-like mitochondrion, with the exception of AOX expression, and remarkably up-regulate proline catabolism and some TCA cycle enzymes.

## Electrons entering the ETC are preferentially channeled to AOX as trypanosomes progress towards metacyclogenesis

To probe the changes in mitochondrial bioenergetics induced by differential expression of key bioenergetic enzymes, we first checked cellular respiration in live cells in medium without any carbon sources. The resting respiration was unchanged in RBP6$^{OE}$ cells (Fig 5A). Adding

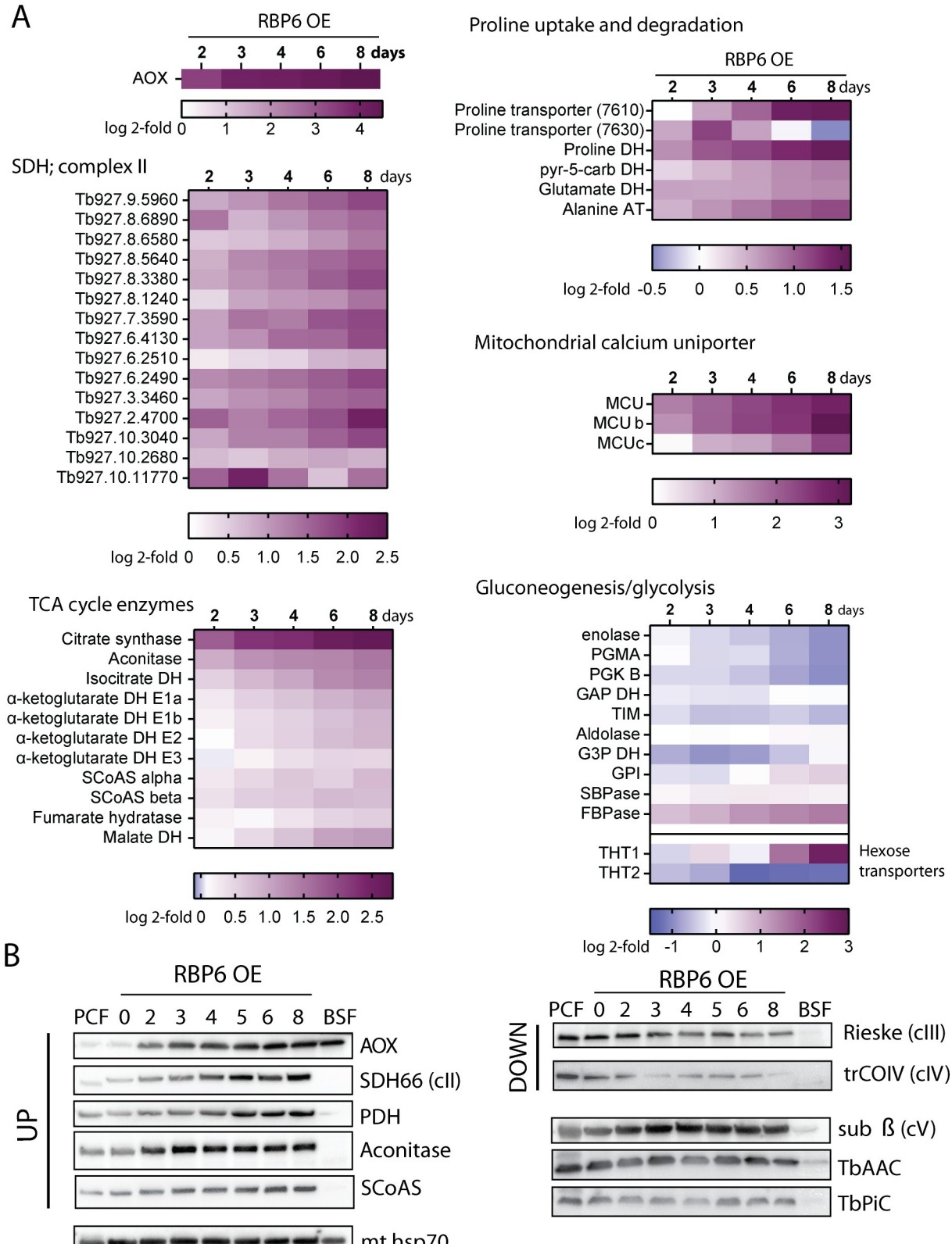

**Fig 3. Major changes in protein abundance during RBP6 overexpression.** (A) Heatmaps showing log2 fold change of average LFQ intensities of selected proteins identified in induced samples compared to uninduced. The color key differs for each map and is always located below the heatmap. (B) Western blot analyses of whole-cell lysates from RBP6^OE cells undergoing differentiation using a panel of various antibodies.

Mitochondrial (mt) hsp70 serves as a loading control because its expression remains constant. AOX, alternative oxidase; AT, aminotransferase; BSF, bloodstream form; DH, dehydrogenase; FBPase, fructose 1,6-bisphosphatase; GAP DH, glyceraldehyde-3-phosphate dehydrogenase; G3P DH, glycerol-3-phosphate dehydrogenase; hsp70, heat shock protein 70; LFQ, label-free quantification; PCF, procyclic form; PDH, pyruvate dehydrogenase; PGK, phosphoglycerate kinase; PGMA, phosphoglycerate mutase; pyr-5-carb DH, pyrroline-5 carboxylate dehydrogenase; RBP6, RNA binding protein 6; SBPase, sedoheptulose 1,7-bisphosphatase; SCoAS, succinyl CoA synthetase; SDH, succinate dehydrogenase; TIM, triose-phosphate isomerase.

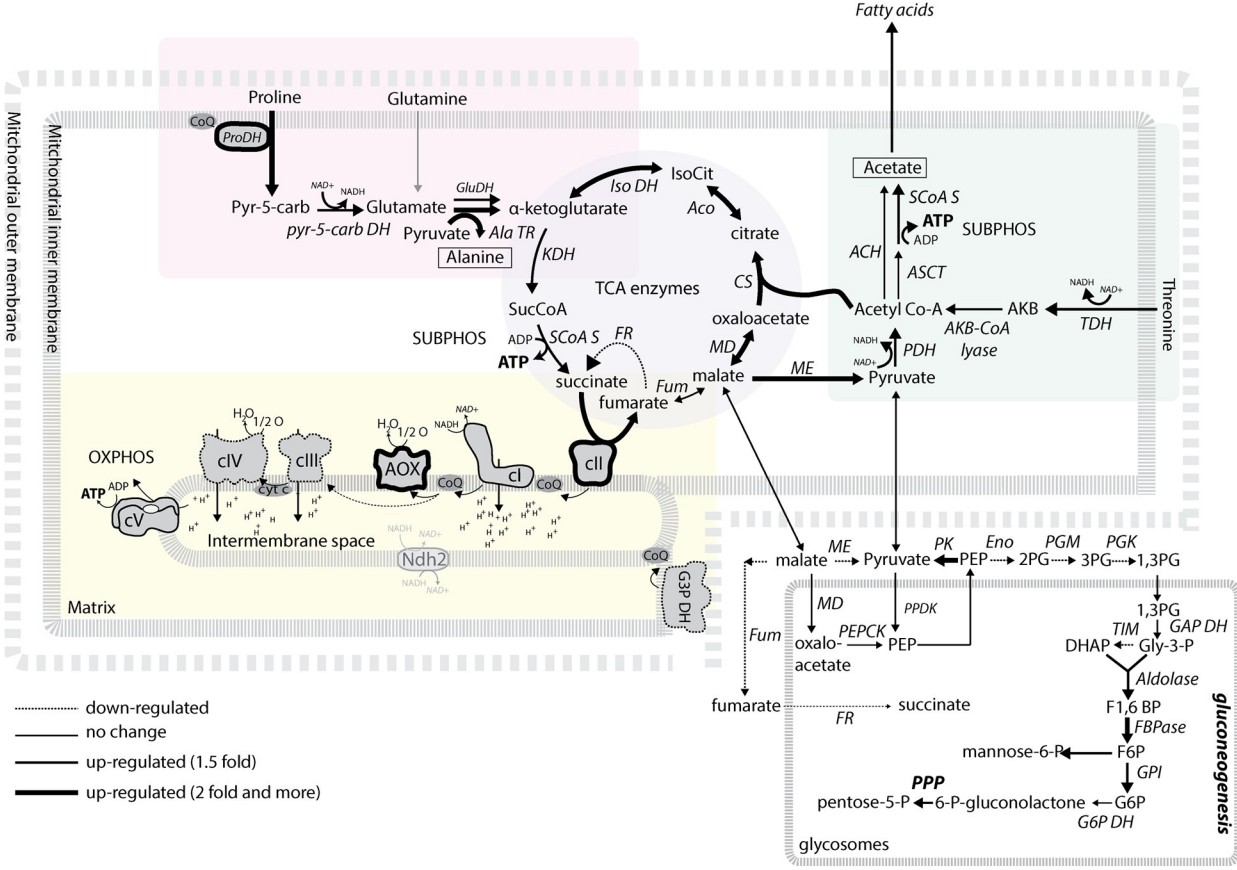

**Fig 4. Schematic representation of changes in selected mitochondrial and glycosomal pathways.** Enzymatic steps are represented by arrows with different thicknesses, depending on the observed abundance change of the respective protein at day 6 upon RBP6 induction. Dashed lines indicate steps that were down-regulated. Alternative dehydrogenase with unresolved orientation is in gray. The metabolic end products are highlighted by black lines. ACH, acetyl-CoA thioesterase; Aco, aconitase; AKB–CoA lyase, 2-amino-3-ketobutyrate Coenzyme A lyase; Ala TR, alanine aminotransferase; AOX, alternative oxidase; ASCT, acetate:succinate CoA-transferase; cI, complex I, NADH:ubiquinone oxidoreductase; cII, succinate dehydrogenase; cIII, complex III, ubiquinol:cytochrome *c* reductase; cIV, complex IV, cytochrome *c* oxidase; cV, $F_oF_1$-ATP synthase; CS, citrate synthase; DHAP, dihydroxyacetone phosphate; Eno, enolase; FBPase, fructose 1,6-bisphosphatase; FR, fumarate reductase; Fum, fumarase; F1,6 BP, fructose 1,6 bisphosphate; F6P, fructose-6-phosphate; GAPDH, glyceraldehyde-3-phosphate dehydrogenase; Glu DH, glutamate dehydrogenase; Gly-3-P, glycerol-3-phosphate; GPI, glucose-6-phosphate isomerase; G3P DH, glycerol-3-phosphate dehydrogenase; G6P, glucose-6-phosphate; G6P DH, glucose-6-phosphate dehydrogenase; Iso DH, isocitrate dehydrogenase; IsoCit, isocitrate; KDH, α-ketoglutarate dehydrogenase; MD, malate dehydrogenase; ME, malic enzyme; OXPHOS, oxidative phosphorylation; PDH, pyruvate dehydrogenase; PEP, phosphoenolpyruvate; PEPCK, phosphoenolpyruvate carboxykinase; PG, phosphoglycerate; PGK, phosphoglycerate kinase; PGM, phosphoglycerate mutase; PK, pyruvate kinase; PPDK, pyruvate, phosphate dikinase; PPP, pentose phosphate pathway; Pro DH, proline dehydrogenase; pyr-5-carb, pyrroline-2-carboxylate; pyr-5-carb DH, pyrroline-5 carboxylate dehydrogenase; RBP6, RNA binding protein 6; SUBPHOS, substrate phosphorylation; SCoAS, succinyl-Coenzyme A synthetase; SucCoA, succinyl-CoA; TDH, threonine dehydrogenase; TIM, triose-phosphate isomerase.

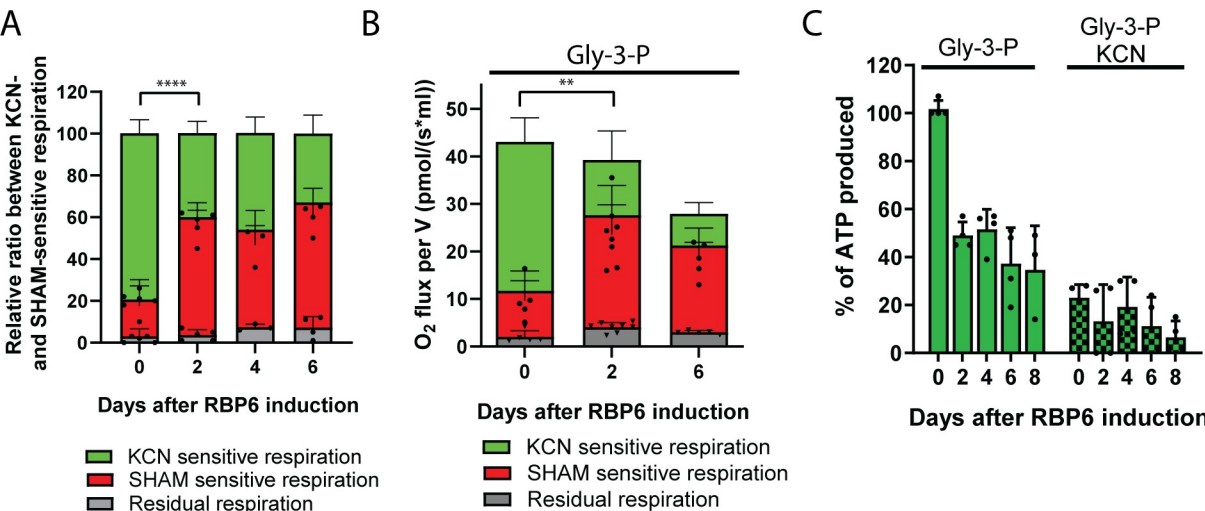

**Fig 5. RBP6<sup>OE</sup> cells respire predominantly via AOX.** (A) The resting respiration of cells undergoing RBP6-induced differentiation was measured using the O2k-oxygraph. The ratio of complex IV–and AOX-mediated respiration was determined using KCN, a potent inhibitor of complex IV, and SHAM, a potent inhibitor of AOX. Individual values shown as dots (mean ± SD, $n$ = 3–5), $^{****}P < 0.0001$. (B) Glycerol-3-phosphate stimulated respiration in live cells. The proportion of complex IV–and AOX-mediated respiration was determined as in (A). Individual values shown as dots (mean ± SD, $n$ = 5–8), $^{**}P < 0.01$. (C) In vitro ATP production was measured in digitonin-extracted mitochondria. The OxPhos pathway was triggered by the addition of ADP and glycerol-3-P. Treatment with 1 mM KCN serves as a control. Individual values shown as dots (mean ± SD, $n$ = 3–4). Underlying data plotted in panels (A), (B), and (C) are provided in S1 Data. AOX, alternative oxidase; Gly-3-P, glycerol-3-phosphate; KCN, potassium cyanide; OxPhos, oxidative phosphorylation; RBP6, RNA binding protein 6; SHAM, salicylhydroxamic acid.

potassium cyanide (KCN), an inhibitor of cytochrome *c* oxidase, showed that this final oxidase is responsible for approximately 80% of oxygen consumption in uninduced cells (Fig 5A, S7 Fig). The increased expression of AOX at day 2 following RBP6 induction caused preferential redirection of the electrons from the canonical cytochrome *c*–mediated pathway to this enzyme. The induced respiration by glycerol 3-phosphate, a substrate for glycerol-3-phosphate dehydrogenase that passes electrons to ubiquinone, also showed preferential oxidation of ubiquinol by AOX during RBP6<sup>OE</sup> (Fig 5B). This rewiring of electron flow resulted in less ATP being produced by OxPhos in digitonin-permeabilized cells in the presence of glycerol 3-phosphate as substrate (Fig 5C), likely because AOX is not linked to the generation of Δψm while electron transfer from ubiquinol to oxygen via complexes III and IV generates Δψm.

To assess if the electrons are preferentially channeled to AOX or this rewiring is a consequence of decreased levels of complexes III and IV, we performed blue-native electrophoresis (BNE) followed by western blotting and activity staining, which is specific for the complexes in question (Fig 6) [11]. At day 2 of RBP6<sup>OE</sup>, no drastic changes in abundance or activity of the complexes III and IV were detected, suggesting that the reduced ubiquinol molecules are preferentially oxidized by AOX, which competes effectively with the cIII/cIV pathway. Furthermore, the in-gel activity assay shows that both complexes III and IV are active throughout the induced development. The western blot analysis of the same samples indicates that the assembly of the complexes is visibly affected at day 6, most likely due to decreased expression of individual subunits (S6 Fig). In agreement with proteomics data illustrating expression profiles of individual subunits of complexes II (Fig 3A) and V (S6 Fig), Fig 6 shows increased activity and abundance of complex II, while complex V remains largely unaffected during the time course of the experiment.

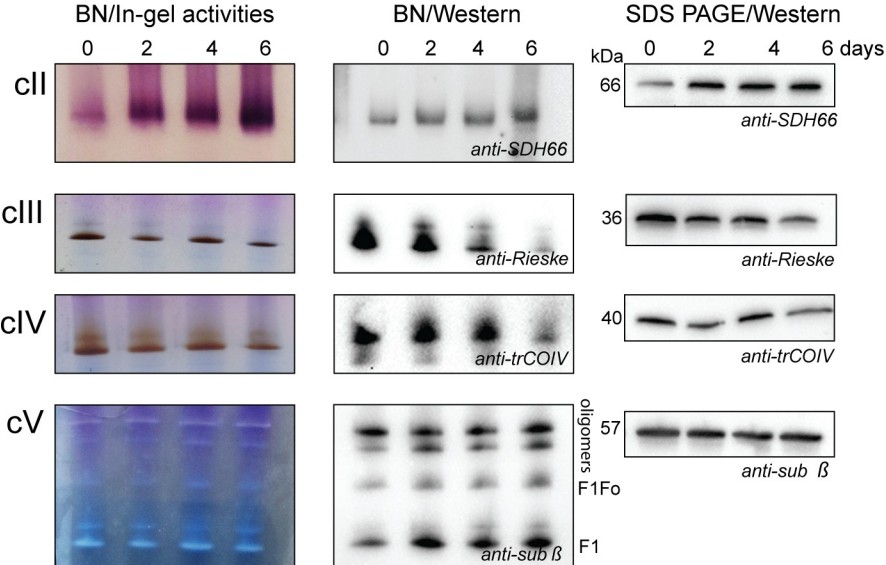

**Fig 6. RBP6^OE-induced changes in levels and activities of ETC complexes and F_oF_1-ATP synthase.** In-gel activity staining and western blot analysis of respiratory complexes II, III, and IV and F_oF_1-ATP synthase (complex V, cV). Mitochondrial preparations were solubilized using dodecyl maltoside and the same amount of the protein samples was separated on NativePAGE 3%–12% Bis-Tris protein gels followed by in-gel activity staining specific for individual complexes (left panels) or by western blot analysis using specific antibodies (middle panels). Mitochondrial lysates were also evaluated by SDS-PAGE and western blot analysis for individual subunits of complexes II, III, IV, and V (right panels). ETC, electron transport chain; RBP6, RNA binding protein 6.

## RBP6 overexpression induces changes in mitochondrial metabolic pathways that lead to increased mitochondrial membrane potential (Δψm)

In our classical view of the PCF mitochondrion, Δψm is maintained by the activity of complexes III and IV [35,36]. Fig 7A shows that Δψm measured in live cells by FACS is significantly increased during the RBP6 induction (Fig 7A). However, in agreement with the electron rewiring to AOX, the detected Δψm was less sensitive to KCN treatment, suggesting that complex IV contributes less to the overall Δψm during in vitro–induced differentiation (Fig 7B). Interestingly, in other eukaryotic systems, during sudden anoxia or induced complex IV dysfunction, the Δψm collapses rapidly and F_oF_1-ATP synthase maintains the Δψm for a short period by reversing its activity [37]. The F_oF_1-ATP synthase hydrolytic activity is regulated by inhibitory peptide 1 (IF1) [38], which is expressed in PCF cells and prevents the reversal of F_oF_1-ATP synthase and thus ATP depletion upon complex IV inhibition [39]. Fig 7C reveals that *T. brucei* IF1 (TbIF1) expression is down-regulated during the RBP6 overexpression. We therefore measured the ability of the mitochondrial proton pumps, complexes III and IV, to generate the Δψm. Utilizing safranine O dye, the RBP6^OE digitonin-permeabilized cells were allowed to establish Δψm in the presence of succinate as the only electron donor. Upon the addition of KCN, the rate of mitochondrial membrane depolarization was a little bit slower in RBP6^OE-induced cells compared to uninduced, while a combined treatment of KCN and oligomycin, an inhibitor of F_oF_1-ATP synthase, depolarized the mitochondrial membranes at the same rate. These results suggest that decreased expression of TbIF1 allows the reversal of F_oF_1-ATP synthase to partially maintain Δψm upon inhibition of complex IV in permeabilized cells (Fig 7D). But as the overall Δψm measured in live cells was not sensitive to oligomycin (Fig 7E), the contribution of F_oF_1-ATP synthase to the total Δψm is most likely negligible. In fact, the oligomycin treatment caused hyperpolarization of mitochondrial inner

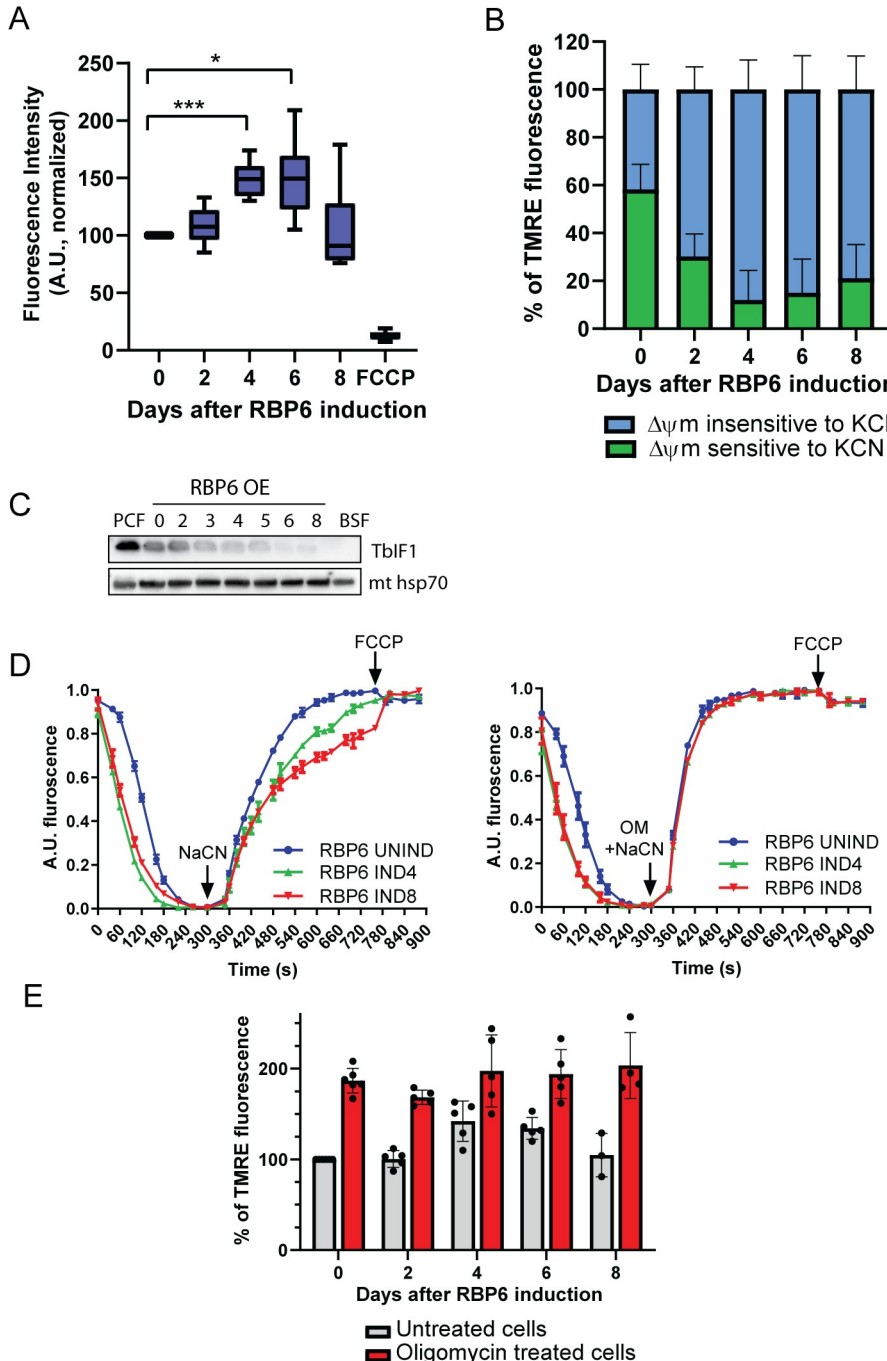

**Fig 7. Mitochondrial membrane potential (Δψm) is increased during RBP6^OE.** (A) The Δψm of RBP6^OE cells post induction was measured by flow cytometry using TMRE. A protonophore FCCP serves as a control for membrane depolarization (mean ± SD, $n = 6–10$) $^*P < 0.05$, $^{***}P < 0.001$. (B) The proportion of Δψm that is generated by complex IV was established by treating the cells with KCN (0.5 mM) in the presence of TMRE for 30 minutes before the analysis. The graph shows a proportion of KCN-sensitive Δψm to the total Δψm measured in each individual sample (mean ± SD, $n = 5$). (C) Western blot analysis of $F_oF_1$-ATPase inhibitory factor TbIF1 during RBP6^OE. Mitochondrial (mt) hsp70 serves as a loading control. (D) The in situ dissipation of the Δψm in response to chemical inhibition of complex IV by 1 mM NaCN was measured using safranine O dye in RBP6^OE uninduced (UNIND) cells and cells induced for 4 and 6 days. The reaction was initiated with digitonin; OM, oligomycin (2.5 μg/mL), and FCCP (5 μM) were added when indicated (mean ± SD, $n = 3$). (E) The Δψm of RBP6^OE cells that were treated (red columns) or not with oligomycin (2.5 μg/mL). Individual values shown as dots (mean ± SD, $n = 3–6$). Underlying data plotted in panels A, B, D, and E are provided in S1 Data. BSF, bloodstream cell; FCCP, carbonyl cyanide-4-phenylhydrazone;

KCN, potassium cyanide; PCF, procyclic cell; RBP6, RNA binding protein 6; TbIF1, *T. brucei* inhibitory peptide 1; TMRE, tetramethyl rhodamine ethyl ester.

membrane in uninduced as well as in RBP6$^{OE}$-induced cells (Fig 7E), implying that F$_o$F$_1$-ATP synthase functions in its forward mode allowing protons to re-enter the mitochondrial matrix to synthesize ATP.

Because the total Δψm is not sensitive to oligomycin and it is less sensitive to KCN, the Δψm during RBP6$^{OE}$ can be partially maintained by complex I, which contributes to Δψm by coupling NADH oxidation with reduction of ubiquinone [40]. The role of complex I in trypanosome mitochondria remains enigmatic because functional studies are hampered by its lack of sensitivity to rotenone, a typical inhibitor of the mitochondrial complex I [41]. Nevertheless, this complex is fully assembled and active, albeit not essential, in procyclic trypanosomes [40]. Proteomics data suggest a significant increase in proline consumption, possibly leading to high levels of reduced NADH and thus higher activity of complex I. This is confirmed by highly enhanced respiration of live cells in the presence of 5 mM proline simulating the growth conditions immediately at day 2 upon RBP6 induction (Fig 8A). The measured increase in oxygen consumption was fully salicylhydroxamic acid (SHAM)-sensitive, suggesting that the majority of electrons is passed to oxygen via AOX (Fig 8A). Similarly, Fig 8B shows an increased succinate-stimulated respiration in digitonin-permeabilized cells, most likely because of elevated abundance and activity of complex II, the succinate dehydrogenase (Fig 6). Interestingly in both cases, we did not observe a dramatic decrease in cytochrome *c*–mediated respiration, suggesting that in the presence of a plentiful carbon source, the canonical pathway maintains its capacity during parasite differentiation. Indeed, in a digitonin-extracted mitochondrial sample, succinate was able to stimulate ATP production via OxPhos (Fig 8C). Because proline consumption is significantly increased during differentiation, one would expect that the oxidation of α-ketoglutarate through the reactions of the TCA cycle will lead to more ATP being produced by substrate phosphorylation pathways, which is integral to the TCA cycle (Fig 4). This pathway uses SCoAS and it is induced by α-ketoglutarate in vitro [42]. Interestingly, α-ketoglutarate did not significantly stimulate ATP production by substrate phosphorylation during the RBP6-induced differentiation (Fig 8D). This observation opens a possibility of α-ketoglutarate entering the reductive branch of the TCA cycle, leading to production of citrate, a phenomenon described in human cells with OxPhos dysfunction [43] (Fig 4). Last but not least, the steady-state cellular levels of ATP were progressively decreased (Fig 8E), indicating a higher need for ATP and consistent with increased ADP/ATP ratio during differentiation (Fig 8F).

## Metabolomic profiling during RBP6$^{OE}$ differentiation

To get further insights into the metabolic changes induced by RBP6 overexpression, we undertook a global metabolomics analysis for the RBP6$^{OE}$ cell line (S6 Table). In agreement with earlier observations, the levels of proline, glutamate, and glutamine were decreased, confirming that proline and glutamine consumption pathways are elevated by day 2 following RBP6 induction (Fig 9A). Possibly due to the high activity of mitochondrial dehydrogenases involved in these pathways, the levels of NADH and thiamine, an essential cofactor for various mitochondrial dehydrogenases, were up-regulated. Accumulation of alanine, the ultimate end product of proline metabolism was detected. Other intracellular amino acids were unchanged or showed non-statistically significant changes with the exception of tyrosine, leucine/isoleucine, and tryptophan (as well as its derivatives 5-hydroxy-L-tryptophan, indole, and quinolinate), the levels of which were up-regulated (Fig 9A). Notably, a comparison of volcano plots

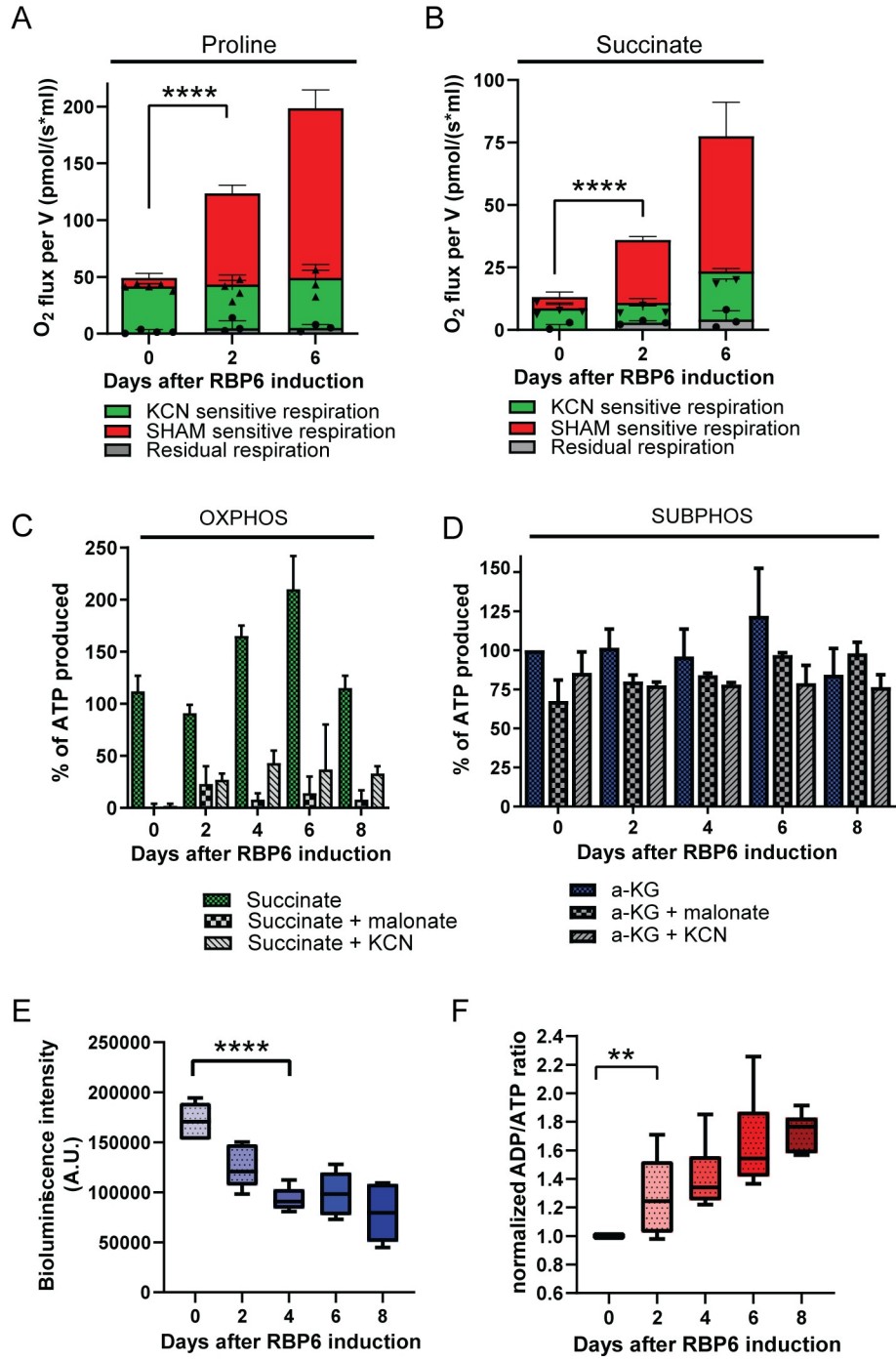

**Fig 8. Changes in cellular respiration and mitochondrial ATP production in RBP6^OE cells.** (A, B) Oxygen consumption rates in the presence of 5 mM proline (A) or 5 mM succinate (B) in live or digitonin-permeabilized cells, respectively. Respiration via AOX was monitored in the presence of KCN (0.5 mM). Individual values shown as dots (means ± SD, $n = 3–5$). $^{****}P < 0.0001$. (C, D) The in vitro ATP production by oxidative or substrate phosphorylation (OXPHOS, SUBPHOS) measured in digitonin-extracted mitochondria from uninduced and RBP6-induced cells. The phosphorylation pathways are triggered by the addition of ADP and by succinate (C) or α-ketoglutarate (a-KG, D). Malonate (mal.) and KCN, specific inhibitors of succinate dehydrogenase and complex IV are used to inhibit ATP production by OXPHOS. The levels of ATP production in mitochondria isolated from uninduced RBP6^OE cells are established as the reference and set to 100% (means ± SD, $n = 2–4$). (E) Cellular ATP content in RBP6^OE cells. (means ± SD, $n = 6$, $^{****}P < 0.0001$). (F) Relative ADP/ATP ratios of RBP6^OE cells. The ADP/ATP ratio in uninduced RBP6^OE cells (between 1.75 and 6.01) is established as a reference and set to 1. In *T. brucei*, ADP/ATP ratio reaches

unusually high levels, as also reported elsewhere [44]. The measured values are shown in S1 Data (means ± SD, $n = 6$–$10$, $**P < 0.01$). Underlying data plotted in panels A, B, C, D, E, and F are provided in S1 Data. AOX, alternative oxidase; KCN, potassium cyanide; OXPHOS, oxidative phosphorylation; RBP6, RNA binding protein 6; SHAM, salicylhydroxamic acid; SUBPHOS, substrate phosphorylation.

generated from data acquired on days 2 and 8 after RBP6 induction suggests an overall accumulation of metabolites (Fig 9B). Gluconeogenesis-related intermediates were mainly unchanged, with the exception of glyceraldehyde 3-phosphate, levels of which were down-regulated (Fig 9C). Nucleotide metabolism was altered with purine-based metabolites being increased. Phosphorylated nucleotides were generally slightly down-regulated (Fig 9D). Among the most striking alterations was an accumulation of several TCA cycle intermediates including malate and citrate (Fig 9E). Interestingly, the molecules crucial for energy storage,

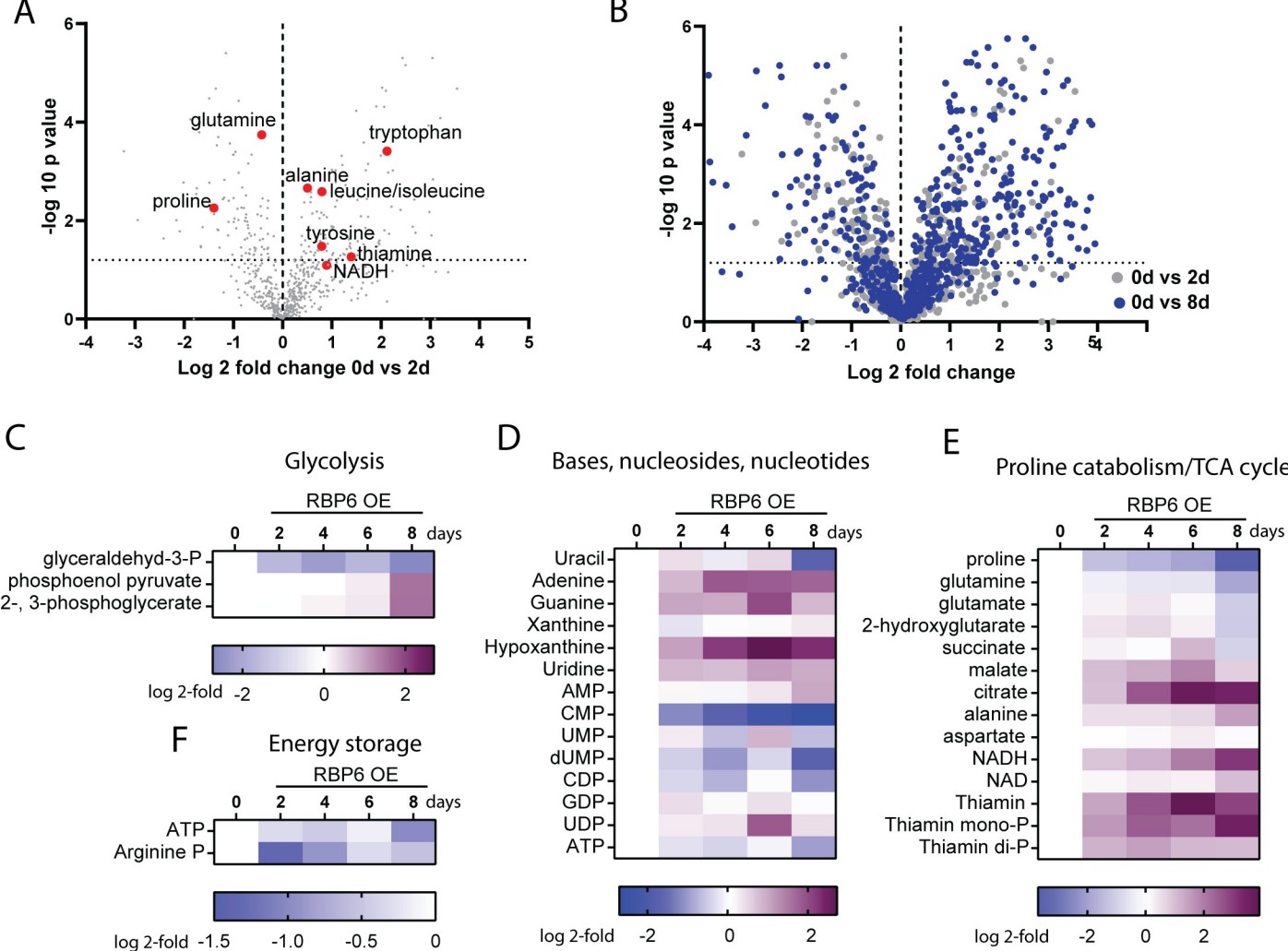

**Fig 9. Metabolomics profiling of RBP6^OE cells.** (A) Volcano plot showing the full metabolome (698 metabolites) analyzed at day 0 and day 2 upon RBP6 induction. Log2 fold change values of the average of mean peak area from quadruplicate experiments are plotted against the respective −log10 transformed *P* values. Few key metabolites are highlighted. (B) Volcano plot showing the full metabolome analyzed at day 2 (gray) and 8 (blue) upon RBP6 induction compared to day 0. Log2 fold change values of the average of mean peak area from quadruplicate experiments are plotted against the respective −log10 transformed *P* values. (D, E, F) Heatmaps showing log2 fold change of average of mean peak area of selected metabolites identified in induced samples compared to uninduced (day 0). The color key differs for each map and is always located below the heatmap. Heatmaps were generated with GraphPad prism 8.2.0. RBP6, RNA binding protein 6; TCA, tricarboxylic acid.

ATP and arginine phosphate, were progressively diminished (Fig 9F). While other changes in metabolite levels across the trypanosome metabolome were detected, no clear trends showing activation or repression of other specific metabolic pathways upon RBP6$^{OE}$ were noted. The full metabolomics dataset can be found in S6 Table and all liquid chromatography–mass spectrometry (LC-MS) data files were deposited in MetaboLights (study identifier MTBLS1390). In summary, the identified changes in the parasite metabolome are consistent with suggested increased activities in mitochondrial dehydrogenases involved in proline consumption and TCA cycle enzymes (Fig 3A and Fig 4).

## Repurposing of the mitochondrion to a ROS-producing signaling organelle

Notably, the levels of glutathione were diminished while the levels of its oxidized product glutathione disulfide were amplified 8-fold at day 8 following RBP6 overexpression (Fig 10A). This was accompanied by a significant 2.2-fold decrease in the levels of the trypanosomatid-specific thiol-based antioxidant ovothiol A at day 2 post expression, and levels were further depleted throughout the time course (S6 Table). The levels of L-cystathionine were also found to be decreased after day 4 following RBP6 overexpression (S6 Table). The latter is an intermediate in the biosynthesis of L-cysteine (not detected in a dataset), and it was reported that enhanced activities of cysteine synthase and cystathionine β-synthase possess a beneficial effect on *Leishmania braziliensis* survival under oxidative stress [45]. The other trypanosomatid-specific derivative of glutathione, trypanothione, was not detected in its reduced form using the LC-MS platform, but its oxidized form, trypanothione disulfide, was annotated in the dataset and was not found to have significantly altered levels across the time course of RBP6$^{OE}$. The increasing abundance of oxidized glutathione is suggestive of mild oxidative stress during developmental progression. We therefore surveyed changes in the expression levels of proteins involved in redox metabolism and, except for putative mitochondrial thioredoxin (Tb927.7.5780), we did not detect any major changes (S8 Fig).

We then analyzed intramitochondrial and intracellular ROS levels (Fig 10B). Interestingly, levels of mitochondrial ROS were elevated immediately after RBP6 induction, while overall cellular ROS were up-regulated only at later time points. ROS molecules, when produced in small concentration, are considered as signaling molecules with the ability to change cell fate and drive cellular differentiation [6]. Excited by the idea that the metabolic repurposing of the mitochondria during the developmental progression leads to the production of signaling molecules, we sought to investigate whether ROS elimination would halt the in vitro–induced differentiation. We took advantage of the fact that *T. brucei* genome lacks catalase, a natural and very potent scavenger of ROS molecules [46]. We introduced the catalase gene from a related organism, *Crithidia fasciculata*, into the *T. brucei* genome under the control of a tetracycline-inducible system. The tetracycline-induced expression and cytosolic localization of the catalase in RBP6$^{OE}$ catalase transduced cells (RBP6$^{OE}$_catalase) were verified by western blot analysis (Fig 10C, inset), and activity was verified visually by a simple assay using live cells, which produced oxygen upon exposure to $H_2O_2$ (S1 Video). An indicator of successful RBP6-induced differentiation is a delay in growth in the first four days post induction, followed by entry into a growth-arrested stationary phase (Fig 10C, red line). Importantly, cells expressing catalase maintained continuous growth (Fig 10C, blue line), and the differentiation kinetics, scored by cell morphology analyses, showed that while the RBP6$^{OE}$_catalase cell line developed epimastigote-like cells after two days, it completely failed to differentiate to metacyclic forms. The fraction of epimastigote-like cells never reached the same proportion as in RBP6$^{OE}$ cells and was eventually overtaken by proliferative procyclics (Fig 10D). Western blot analysis showed that both cell lines expressed RBP6 protein to similar levels (Fig 10E). The major differences were

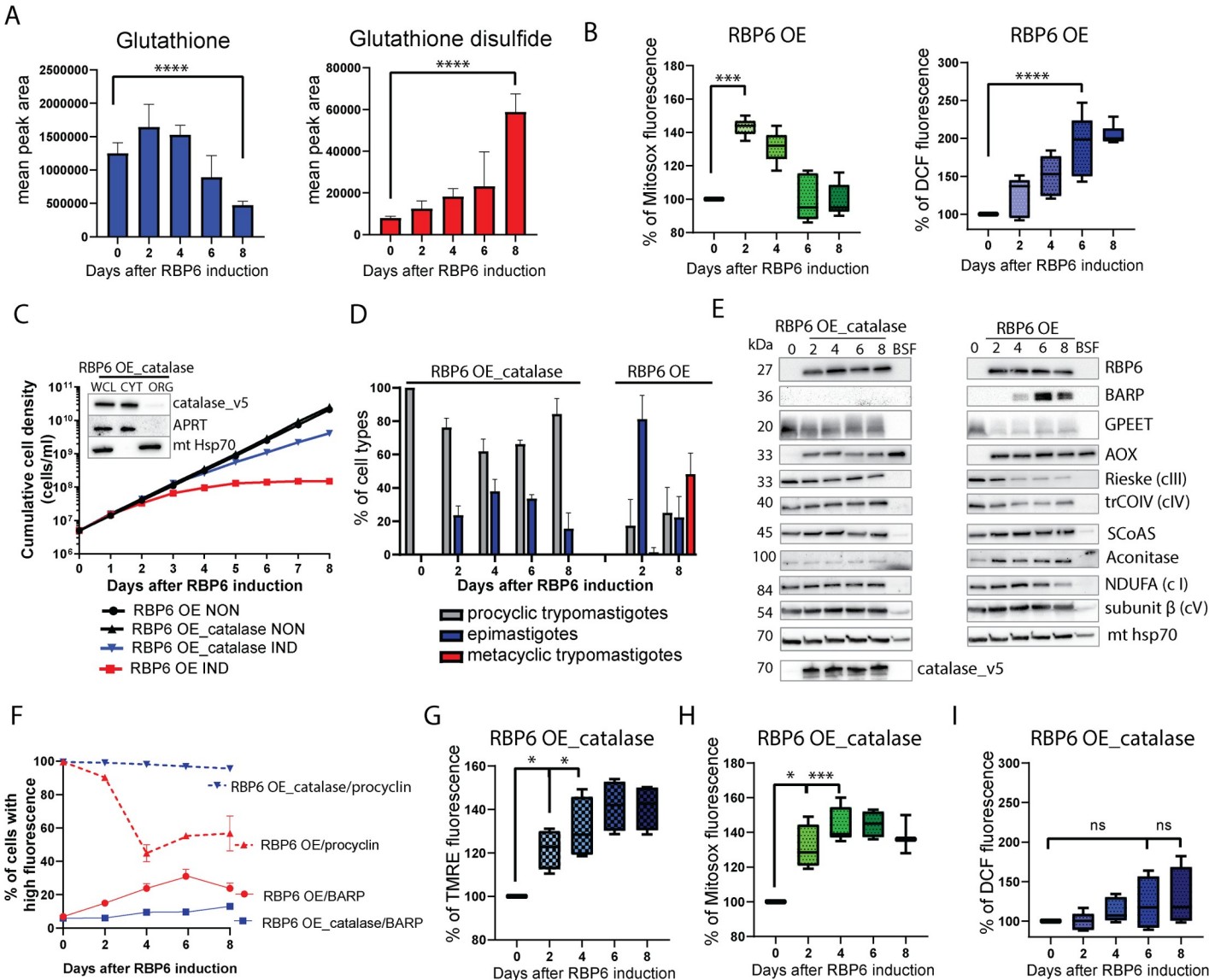

**Fig 10. Elevated ROS generated during the differentiation of *T. brucei* are crucial for driving forward RBP6-induced differentiation.** (A) Metabolites glutathione and glutathione disulfide as detected by LC-MS analyses. The size of the bars represents the total abundance of the metabolite (mean ± SD, $n = 4$, ***$P < 0.001$). (B) Mitochondrial and cellular ROS detection reagents (MitoSox and H$_2$DCFDA, respectively) were quantified by FACS (means ± SD, $n = 5$, ***$P < 0.001$, ****$P < 0.0001$). (C) Representative growth curves of RBP6$^{OE}$ and RBP6$^{OE}$_catalase cells induced by tetracycline. Total number of experiments $n = 5$. The inset shows subcellular localization of v5-tagged catalase in RBP6$^{OE}$_catalase cells induced for 48 hours. Immunoblots were labeled with anti-v5, anti-adenosine phosphoribosyl transferase (APRT), and anti-mt hsp70 antibodies to visualize catalase, cytosolic APRT, and mitochondrial localized hsp70, respectively. (D) Time line for the appearance of epimastigotes and metacyclic cells upon induction of RBP6$^{OE}$ and RBP6$^{OE}$_catalase. Total number of experiments $n = 3$. (E) Western blot analysis of whole-cell lysates from RBP6$^{OE}$ and RBP6$^{OE}$_catalase cells using available antibodies. (F) FACS analyses of RBP6$^{OE}$ and RBP6$^{OE}$_catalase cells treated with 5 mM bathophenanthroline disulphonic acid (BPS) and labeled with polyclonal anti-BARP and anti-procyclin antibodies. (G) The Δψm of RBP6$^{OE}$_catalase cells post induction measured by flow cytometry using TMRE (means ± SD, $n = 4$), *$P < 0.05$. (H) Mitochondrial ROS detection reagent MitoSox in RBP6$^{OE}$_catalase cells quantified by FACS (means ± SD, $n = 5$), *$P < 0.05$, ***$P < 0.001$. (I) Cellular ROS detection reagent H$_2$DCFDA in RBP6$^{OE}$_catalase cells quantified by FACS (means ± SD, $n = 5$). Underlying data plotted in panels A, B, C, D, F, G, H, and I are provided in S1 Data. AOX, alternative oxidase; APRT, adenine phosphoribosyl transferase; BARP, brucei alanine-rich protein; CYT, cytosol; hsp70, heat shock protein 70; H$_2$DCFDA, 2′,7′-dichlorofluorescin diacetate; LC-MS, liquid chromatography–mass spectrometry; ns, statistically not significant; ORG, organellar fraction; RBP6, RNA binding protein 6; ROS, reactive oxygen species; SCoAS, succinyl Co-A synthetase; TMRE, tetramethyl rhodamine ethyl ester; WCL, whole-cell lysate.

seen in the expression of surface glycoproteins, GPEET and BARP. Overexpression of catalase during the RBP6$^{OE}$ interfered with the programmed destabilization of GPEET, whose expression in vivo is controlled by the activity of mitochondrial enzymes [47]. BARP protein, a

hallmark molecule for the surface of salivary gland epimastigotes, however, was not detected in RBP6$^{OE}$_catalase cells (Fig 10E). These results suggest that the expression of catalase blocks the differentiation earlier than the transition from BARP-positive mature epimastigotes to metacyclic forms. The steady expression of the procyclin-containing coat in RBP6$^{OE}$_catalase cells was confirmed by flow cytometry analysis (Fig 10F). The changes in the expression of mitochondrial proteins were also detected, when RBP6$^{OE}$_catalase cells are compared to RBP6$^{OE}$ cells. In RBP6$^{OE}$_catalase cells, in contrast to RBP6$^{OE}$ cells, the expression of trCOIV and Rieske was not decreased and we did not detect changes in the expression of SCoAS and aconitase (Fig 10E). Similarly to the RBP6$^{OE}$ cells, the $\Delta\psi m$ was elevated in the RBP6$^{OE}$_catalase cells (Fig 10G), as was mitochondrial ROS production (Fig 10H). Cytosolic catalase, however, scavenged cytosolic ROS because no statistically significant increase in ROS measured by 2′,7′-dichlorofluorescein (DCF) was detected (Fig 10I).

In summary, our data suggest that RBP6 expression induces differentiation to epimastigote cells, accompanied by significant changes in mitochondrial metabolism. These alterations lead to increased production of ROS molecules, the levels of which seem to be important for efficient differentiation towards metacyclogenesis. Our results provide insights into the mechanisms of the parasite's mitochondrial rewiring and reinforce the emerging concept that mitochondria act as signaling organelles through the release of ROS to drive cellular differentiation.

## Discussion

Our knowledge of molecular mechanisms driving metabolic rewiring of the *T. brucei* mitochondrion from a fully competent organelle capable of OxPhos to its metabolically reduced ATP-consuming version is limited. Until recently, investigations of the developmental program inside of the tsetse [19] required fly infections and laborious dissections [48]. This void has been partially overcome by introducing an in vitro differentiation system based on induced overexpression of the RBP6 protein [25]. Compared to this study, we achieved faster differentiation (higher levels of metacyclics at day 6 versus day 10) and higher efficacy (50% of metacyclics versus 32%) by growing the RBP6$^{OE}$ cells in SDM-80 medium supplemented with a fetal bovine serum (FBS), which contains proline, not glucose, as the energy source. To prevent uptake of residual glucose molecules from the FBS, the RBP6$^{OE}$ cells were grown in the presence of N-acetyl glucosamine (a non-transported glucose analog that binds the transporter). Because the efficacy of differentiation correlates with the expression of RBP6 [25], our results suggest that the absence of glucose increases RBP6 expression in vitro, and possibly RBP6 gene expression is controlled by environmental stimuli, as reported for other genes such as GPEET [27]. Proline and glutamine are the major amino acids in the haemolymph of the tsetse fly. In PCF mitochondria, proline is oxidized by several enzymatic reactions to α-ketoglutarate, a key anaplerotic substrate of the TCA cycle, preceding further metabolism to succinate and alanine [49]. Upon the RBP6 induction, we detected lower levels of proline and glutamate, which may suggest a higher consumption rate, with a concomitant increase in expression of enzymes involved in their catabolism and in the levels of the catabolic end product, alanine. Given that alanine is the predominant building block of the parasite's dynamic surface coat composed of BARP, the parasite might be satisfying its higher need for this amino acid. Interestingly, low alanine levels were detected in RNA interference-silenced Δ-pyrroline-5-carboxylate dehydrogenase (TbP5CDH) cells, which were incompetent in the efficient establishment of infection in the tsetse fly [50]. Higher activities of mitochondrial dehydrogenases together with the α-ketoglutarate entry to the TCA cycle should boost mitochondrial ATP production to support a higher need for cellular energy currency, which might be required for the energy-costly

reorganization of the kinetoplast from the posterior to the anterior side and back during the differentiation to epimastigotes and metacyclics [22]. In agreement with the proposed higher energy demand, we measured decreased steady-state levels of cellular ATP and arginine-phosphate with increased levels of ADP/ATP ratio.

Enzymatic activities of all TCA cycle enzymes have been detected in PCF trypanosomes [51], but direct experimental evidence revealed that the parasite does not use the full cycle but rather its partial reactions [52]. The metabolite α-ketoglutarate can be metabolized via pyruvate to alanine or acetate or just partially oxidized to malate, which is then diverted to feed gluconeogenesis, a process essential for in vivo development in the tsetse fly [53]. Four canonical TCA cycle enzymes, malate dehydrogenase, CS, aconitase, and IDH seem to have no metabolic role in trypanosomes grown either in glucose or proline/threonine [54,55]. Strikingly, all of these enzymes were strongly up-regulated during the RBP6 overexpression, with CS being the most affected (increased in expression by 3.5 to 6.5 times during the differentiation). Another remarkable observation was an up-regulation of a putative mitochondrial citrate transporter (Tb927.9.4310; S8 Fig) and high levels of intracellular citrate. This intermediate TCA cycle metabolite can be produced from either threonine-derived acetyl-CoA or by reductive carboxylation of α-ketoglutarate involving a reversal of NADP-dependent IDH and aconitase. In mammalian cells, the bifurcation of the TCA cycle to its reductive branch responds to a high α-ketoglutarate/citrate ratio and is crucial for the provision of lipogenic citrate during hypoxia, aglycemia, or when mitochondrial respiration is impaired. Citrate is then transported to the cytosol, where it is converted to lipogenic acetyl-CoA by the action of cytosolic ATP-dependent citrate lyase (ACL) [43,56]. *T. brucei* does not synthesize lipids from citrate-derived acetyl-CoA; instead, the parasite uses mitochondrial acetate for further lipogenesis and cholesterolgenesis [57]. Therefore, it is unlikely that citrate accumulation would yield a higher fatty acid synthesis. Instead, citrate can be converted to isocitrate by action of cytosolic aconitase [58] and further oxidized to α-ketoglutarate by glycosomal NAD/NADP-dependent IDH [59] to maintain NAD(P)H redox balance. Reduced NADPH is also critical for oxidative stress defense.

ROS (e.g., superoxide and $H_2O_2$) are important regulators of cellular homeostasis and play an essential role in cellular stress signaling, cell survival, differentiation, proliferation, and oncogenic transformation [5,6,60]. The ETC is thought to be the main source of mitochondrial ROS in a process that can be triggered by increased respiratory rate, changes in $\Delta\psi$m, and dysfunctional ETC complexes. ETC complexes I, II, and III contain sites wherein electrons can prematurely reduce oxygen, resulting in the formation of superoxide, which can be further converted to $H_2O_2$ by superoxide dismutases (SODs). In RBP6[OE] cells, a significant increase in mitochondrial ROS production coincides with elevated expression of AOX and with rechanneling of electrons entering ETC to this oxidase. This observation, interestingly, contradicts the available literature on AOX expression in other systems. For example, the introduction of *Ciona intestinalis* AOX into mammalian cells with severe mitochondrial dysfunction minimized ROS production by bypassing complexes III and IV [61,62], re-activating the electron flow and thereby maintaining redox homeostasis for TCA cycle activity [63].

As electrons are redirected away from complex III, during RBP6[OE], complex I is the prominent candidate for the observed ROS production. Although the abundance of this complex was not affected by RBP6 overexpression, its activity should be increased because of the elevated levels of NADH-producing enzymes involved in proline oxidation. Complex I can produce ROS at the entrance flavin mononucleotide site in a manner dependent on the $NAD^+$/ NADH ratio, independently of the coenzyme Q (CoQ) pool status [64], or by reverse electron transfer (RET) driven by succinate oxidation, substantial $\Delta\psi$m, and highly reduced CoQ pool [65]. While during the in vitro–induced differentiation, we observed conditions that could

support RET, without further studies defining the CoQ redox poise, the role of complex I in ROS production during RBP6[OE] will remain elusive.

ROS molecules also play an important role in the differentiation of the *T. brucei*–related parasite *Leishmania* [66]. Following infection of mammalian macrophages, *Leishmania* promastigotes differentiate to amastigotes. Intriguingly, ROS-generating drug menadione or $H_2O_2$ alone triggered promastigote differentiation to fully infective amastigote [66]. Here, we show that ROS molecules produced during the RBP6 overexpression are important for the completion of the developmental program, as overexpression of cytosolic catalase hindered the in vitro–induced differentiation. In vivo, catalase expression also impeded the ability of *T. brucei* to establish infection in the midgut of the tsetse fly [67]. The ROS-induced signal transduction pathway likely takes place in the cytosol, because only the cytosolic ROS levels were decreased to the original values upon the catalase induction. $H_2O_2$ molecules are prominent candidates for ROS signaling as they are long-lived, membrane-permeable, and induce reversible oxidation of thiols present in redox-sensitive proteins [68]. At the cellular level, in yeast, mitochondrion-induced $H_2O_2$ signal was shown to attenuate global protein synthesis by modulating the redox status of proteins involved in translation [69]. Intriguingly, translational attenuation is a hallmark of *T. brucei* metacyclic cells, which are quiescent (arrested in G1/G0 phase [26]). Multiple kinases and phosphatases are also susceptible to cysteine oxidation, with activity controlled by redox signals. ROS-induced activation of AMP-activated kinase alpha, for example, is important in inducing quiescence in stumpy BSF trypanosomes [70], and inactivation of *T. brucei* tyrosine phosphatase TbPTP1 is important for in vitro–induced differentiation of the bloodstream stumpy forms to PCF [71].

In summary, we have provided a global transcriptomic, proteomic, and metabolomic study of *T. brucei* cells undergoing an 8-day-long in vitro differentiation to metacyclic cells. Even though RBP6[OE] is genetically induced and thus an artificial route to the differentiation of the midgut PCF trypomastigote, our–omics and functional data constitute a unique set of resources to facilitate further interrogation of intrinsic and extrinsic signaling pathways to gain deeper insights into the differentiation processes underlying not only rewiring of mitochondrial metabolism and its consequences but also changes in gene expression of surface proteins, kDNA repositioning, and mitochondrial cristae remodeling.

## Material and methods

### RNA preparation, RNA-Seq, read processing, and data analysis

Total RNA was isolated from *T. brucei* at different stages ($1 \times 10^8$ cells/replicate) using the miRNeasy Kit (Qiagen, Germany) according to the manufacturer's protocol. An additional DNase1 digestion step was performed to ensure that the samples were not contaminated with genomic DNA. RNA purity was assessed using the Agilent 2100 Bioanalyzer (Agilent Technologies, Santa Clara, CA). Next generation sequencing (NGS) library prep was performed with TruSeq Strand-Specific mRNA Library Prep with PolyA-Selection following Illuminas standard protocol. Libraries were profiled using the High Sensitivity DNA Kit on a 2100 Bioanalyzer and quantified using the Qubit dsDNA HS Assay Kit, in a Qubit 2.0 Fluorometer (Life technologies). Libraries were sequenced on an Illumina NextSeq 500 in the Genomics Core Facility at the Institute of Molecular Biology, Mainz, Germany. The RNA-Seq measurement yielded on average 11.1 M single reads of 75 nt per sample. We assessed the quality of the sequenced reads with fastqc [72] and dupRadar (https://doi.org/10.1186/s12859-016-1276-2). Spliced-leader sequences (CAATATAGTACAGAAACTGTTCTAATAATAGCGTTAGTT) were removed from the reads using cutadapt (https://doi.org/10.14806/ej.17.1.200) and then mapped to the *T. brucei* 11 megabase chromosomes (TriTrypDB version 36) using STAR

version 2.5.2b (https://doi.org/10.1093/bioinformatics/bts635), allowing up to 2 mismatches, a minimum intron length of 21, and keeping only uniquely aligned reads (on average, 75.1% of all reads). We then counted reads per gene using featureCounts (https://doi.org/10.1093/bioinformatics/btt656) from the subread package version 1.5.1 with default parameters and using the gene models provided by TriTrypDB version 36. For the differential expression analysis, we used R version 3.4.3 (http://www.R-project.org/) and DESeq2 version 1.18.1 (https://doi.org/10.1186/s13059-014-0550-8) to normalize, transform, and model the data. The counts were fitted to a Negative Binomial generalized linear model (GLM), and the Wald significance test was used to determine the differentially expressed genes between control and knockdown samples. Finally, RPKM values were calculated per gene using the library size-normalized FPM (robust counts per million mapped fragments) values from DESeq2. We applied automatic independent filtering to avoid testing genes, which were poor candidates of being differentially expressed (maximizes the number of adjusted *P* values less than alpha = 0.1). Differentially expressed mRNAs were identified using a threshold of Benjamini-Hochberg–corrected *P* values <0.05. GO enrichment analyses were performed using GO Term annotations TriTrypDB-36_TbruceiLister427_GO.gaf from TriTrypDB version 36 and Fisher's exact test. For the comparison to the transcriptome data of the previously published dataset of procyclic and metacyclic forms of *T. brucei* [26,31], the respective RNA-Seq raw data files were downloaded from SRA (Bioproject PRJNA381952) and processed exactly as described above for our data.

## Mass spectrometry sample preparation, MS measurement, and proteomics data analysis

*T. brucei* at different stages ($1 \times 10^8$ cells/replicate) were washed three times in 10 mL of phosphate-buffered saline (PBS) and lysed in 6% sodium dodecyl sulfate (SDS), 300 mM DTT, and 150 mM Tris-HCl (pH 6.8), 30% glycerol, and 0.02% Bromophenol Blue. Samples were loaded on a NOVEX NuPage 4%–12% gradient gel (Thermo Fisher Scientific, Waltham, MA), run for 10 minutes at 180 V, and stained with Coommassie G250. Each lane was cut and the minced gel pieces were transferred to an Eppendorf tube for destaining with 50% ethanol/50 mM ABC buffer pH 8.0. The gel pieces were dried and subsequently reduced (10 mM DTT/50 mM ABC buffer pH 8.0), alkylated (55 mM iodoacetamide/50 mM ABC buffer pH 8.0), and digested with 1 μg trypsin overnight at 37˚C. The tryptic peptides were eluted from the gel pieces with pure acetonitrile and stored on a StageTip [73].

The proteomic measurement was performed on a Q Exactive Plus mass spectrometer (Thermo Fisher Scientific, Waltham, MA) with an online-mounted C18-packed capillary column (New Objective, Woburn, MA) by eluting along a 225-minute gradient of 2% to 40% acetonitrile using an EasyLC 1000 uHPLC system (Thermo Fisher Scientific, Waltham, MA). The mass spectrometer was operated with a top10 data-dependent acquisition (DDA) mode.

Data analysis was performed in MaxQuant [74] version 1.5.2.8 using the tritrypDB-43_TbruceiLister427_2018_AnnotatedProteins database (16,869 entries) and standard settings, except activating the match between run feature and the label-free quantification (LFQ) algorithm. Protein groups marked as contaminants, reverse entries, and only identified by site were removed prior to bioinformatics analysis, as well as protein groups with less than 2 peptides (minimum 1 unique). Additional information like gene names and descriptions were extracted from the fasta header and attached to the individual protein groups. Additionally, we identified the best ortholog to Tb927 by using the inparanoid algorithm [75]. Imputation of missing values was performed using a beta distribution within 0.2 and 2.5 percentile of measured values for individual replicates separately. PCA plot was created using R package ggbiplot-0.55; the heatmap was produced by heatmap.2 command from gplots-3.0.1.1 package.

Clustering was performed using the pamk function from fpc-2.2–1 package with "usepam" deactivated and a "krange" between 5 and 9. The algorithm performs a partitioning around medoids clustering of large datasets, with the optimal number of clusters estimated by optimum average silhouette width. GO enrichment was performed using Fisher's Exact Test, and *P* values were corrected using the Benjamini-Hochberg method.

## Trypanosome culture conditions and generation of cell lines

*T. brucei* PCF cells strains 29.13, transgenic for T7 RNA polymerase and the tetracycline repressor [76], were grown in vitro at 27˚C in SDM-80 medium containing hemin (7.5 mg/mL) and 10% FBS. The pLew100v5 vector for RBP6 expression (a kind gift from Prof. Tschudi) was linearized with NotI enzyme and transfected into the 29.13 cell line. RBP6$^{OE}$ cells were adapted to grow in SDM-80 medium containing no glucose and further supplemented with 50 mM N-acetyl glucose-amine to block uptake of residual glucose molecules from 10% FBS. The induction of ectopically expressed RBP6 protein was triggered by the addition of 10 μg/mL of tetracycline into the medium. Cell densities were measured using the Z2 Cell Counter (Beckman Coulter, Brea, CA). Throughout the analyses, cells were maintained in the exponential mid-log growth phase (between $2 \times 10^6$ and $1 \times 10^7$ cells/mL). The RBP6$^{OE}$_catalase cell line was generated using pT7v5 plasmid containing a *C. fasciculata* catalase ORF sequence (cCAT TriTrypDB gene ID = CFAC1_250006200) [67]. The pT7v5_catalase plasmid was linearized with NotI and transfected into RBP6$^{OE}$ cells. The activity of the catalase was detected using a simple visual activity test. A total of $5 \times 10^7$ parasites were resuspended in 100 μL of PBS and placed on a microscopic slide. A volume of 20 μL of 3% $H_2O_2$ was added to the cells, mixed, and the formation of oxygen (bubbles formation) was monitored by eye.

## Isolation of mitochondrial vesicles, BNE, and high-resolution clear-native PAGE

BNE of mitochondria lysed with dodecylmaltoside, followed by in-gel activity staining, was adapted from published protocols [11]. Briefly, the mitochondrial vesicles from $5 \times 10^8$ cells were isolated by hypotonic cell lysis, and mitochondria were resuspended in a buffer (750 mM aminocaproic acid, 50 mM Bis-Tris, 0.5 mM EDTA pH 7.0, supplemented with the complete EDTA-free protease inhibitor cocktail [Roche, Basel, Switzerland]) and lysed for one hour on ice with 2% dodecylmaltosid. The samples were spun down at 16,000*g* for 30 minutes and the cleared lysate protein concentrations were determined by a BCA assay. Mitochondrial lysate (20 μg) was mixed with a loading dye (50 mM ACA, 0.5% [w/v] Coomassie Brilliant Blue G-250). After electrophoresis (3 hours, 150 V, 4˚C), the resolved mitochondrial lysates were transferred onto a PVDF membrane and probed with selected antibodies, or the native gels were directly used for in-gel activity staining of the respiratory complexes. Specific staining of complex III was achieved by incubating the gel in complex III assay buffer (1 mg/mL of 3,3′ diaminobenzidine, 50 mM sodium phosphate pH 7.4, 75 mg/mL sucrose) by slow agitation overnight. Complex IV staining was achieved using a reaction buffer (50 mM phosphate buffer pH 7.4, 1 mg/mL 3,3′diaminobenzidine, 24 U/mL catalase, 1 mg/mL cytochrome *c*, 75 mg/mL sucrose) overnight. Complex V was visualized using ATPase reaction buffer (35 mM Tris-HCl pH 8.0, 270 mM glycine, 19 mM MgSO$_4$, 0.3% [w/v] Pb(NO$_3$)$_2$, 11 mM ATP) for overnight incubation by slow agitation. Mitochondrial samples for complex II detection were treated with loading dye (0.1% Ponceau-S, 50% glycerol) and they were run on 3%–12% gradient high-resolution clear-native gels (hrCNE). Specific staining was achieved by incubating the gel in staining solution: 5 mM Tris HCl 7.4, 20 mM sodium succinate, 0.2 mM phenazine methasulfate, and 2.5 mg/mL nitrotetrazolium blue chloride.

## SDS-PAGE and western blot

Protein samples were separated on SDS-PAGE, blotted onto a PVDF membrane (Thermo Fisher Scientific, Waltham, MA), and probed with the appropriate monoclonal antibody (mAb) or polyclonal antibody (pAb). This was followed by incubation with a secondary HRP-conjugated anti-rabbit or anti-mouse antibody (1:2,000, BioRad). Proteins were visualized using the Pierce ECL system on a ChemiDoc instrument ((BioRad, Hercules, CA)). The PageRuler prestained protein standard (Fermentas, Vilnius, Lithuania) was used to determine the size of the detected bands. Several antibodies (anti-Rieske, anti-trCOIV, anti-SCoAS, anti-aconitase, and anti-NDUFA) were prepared for the purpose of this study. The open reading frames of the respective genes without their predicted mitochondrial localization signal were cloned into the *E. coli* expression plasmid, pSKB3. The proteins were overexpressed in BL21 *Escherichia coli* cells and purified under native or denatured conditions. Antigens were sent to Davids Biotechnologie (Regensburg, Germany) for pAb production. Primary antibodies used in this study were as follows: mAb anti-mitochondrial hsp70 (1:2,000) [77], pAb anti-RBP6 (1:5,000, a generous gift from Prof. Tschudi), pAb anti-BARP and GPEET (1:2,000, 1:1,000, respectively, a generous gift from Prof. Roditi), mAb AOX (1:100) (a generous gift from Prof. Chaudhuri), mAb41-PDH (1:100) [77], pAb anti-Rieske (1:1,000, commercially produced for the purpose of this study), pAb trCoIV (1:1,000, commercially produced for the purpose of this study), pAb anti-subunit beta, $F_1$-ATPase (1:2,000) [78], pAb anti-SCoAS (1:1,000, commercially produced for the purpose of this study), pAb anti-aconitase (1:2,000, commercially produced for the purpose of this study), pAb anti-AAC (1:2,000) [79], anti-SDH1 [80], pAb anti-TbIF1 [39], pAb anti-PiC (1:500) [79], pAb anti-NDUFA (1:1,000, commercially produced for the purpose of this study).

## Cellular ROS, mitochondrial ROS, and mitochondrial membrane potential (Δψm) measurements

The cellular and mitochondrial ROS levels were determined using the 2′,7′-dichlorofluorescein diacetate ($H_2$DCFHDA) and MitoSOX Red Mitochondrial Superoxide dyes (Thermo Fisher Scientific Waltham, MA), respectively. Cells in the exponential growth phase were treated with 10 μM $H_2$DCFHDA or with 5 μM MitoSOX for 30 minutes at 27°C. A total of $1 \times 10^7$ cells were pelleted (1,300*g*, 10 minutes, RT), washed with 1 mL of PBS (pH 7.4), resuspended in 2 mL of PBS, and immediately analyzed by flow cytometry (BD FACS Canto II Instrument, BD Biosciences, San Jose, CA). The Δψm was determined using the red-fluorescent stain tetra-methylrhodamine ethyl ester TMRE (Thermo Fisher Scientific Waltham, MA). Cells in the exponential growth phase were stained with 60 nM of the dye for 30 minutes at 27°C. Cells were pelleted (1,300*g*, 10 minutes, RT), resuspended in 2 mL of PBS (pH 7.4), and immediately analyzed by flow cytometry (BD FACS Canto II Instrument). Treatment with the protonophore FCCP (20 μM) was used as a control for mitochondrial membrane depolarization. To evaluate the effect of oligomycin and KCN on Δψm, the cells were incubated with 2.5 ug/mL of oligomycin or 0.5 mM of KCN in the presence of TMRE for 30 minutes at 27°C. For all samples, 10,000 events were collected. Data were evaluated using BD FACSDiva software (BD Biosciences, San Jose, CA).

## In situ Δψm measurement

Estimation of the Δψm in situ was performed spectrofluorometrically using the indicating dye safranine O (Sigma-Aldrich, St. Louis, MO). *T. brucei* PCF cells ($2 \times 10^7$ cells/mL) were resuspended in a reaction buffer containing the following: 200 mM sucrose, 10 mM HEPES-Na

(pH 7.0), 2 mM succinate, 1 mM MgCl$_2$, and 1 mM EGTA. The reaction was induced with digitonin (40 μM), while NaCN (1 mM) and FCCP (5 μM) were injected at specific time points throughout the assay. Changes in the amount of fluorescence over time were detected on an Infinite M200 microplate reader (TECAN) (excitation = 496 nm; emission = 586 nm). Values were normalized according to the following equation: normalized ($E_i$) = $E_i - E_{min}/E_{max} - E_{min}$ ($E_{min}$ – the minimum value for variable E, $E_{max}$ – the maximum value for variable E).

## Measurement of oxygen consumption

The oxygen consumption rate was assessed at 27˚C using $2 \times 10^7$ cells per Oroboros O2K oxygraph chamber. The endogenous respiration of cells, as well as the substrate-induced respiration, were measured in MiR05 medium (Oroboros, Innsbruck, Austria). The following respiratory substrates were used: 10 mM succinate, 5 mM proline, or 10 mM glycerol-3-phosphate. The SHAM-sensitive respiration was determined by injecting 250 μM SHAM, while the KCN sensitive respiration was assessed with 1 mM KCN. If needed, the cells were permeabilized with 4 ug of digitonin (Sigma-Aldrich, St. Louis, MO).

## ATP production assay

ATP production was measured as described [42]. Briefly, crude mitochondrial fractions from the RBP6$^{OE}$ cells were obtained by digitonin extraction. ATP production in these samples was induced by the addition of 5 mM of indicated substrates (succinate, α-ketoglutarate, and glycerol 3-phosphate) and 67 μM ADP. The mitochondrial preparations were preincubated for 10 minutes on ice with the inhibitor malonate (6.7 mM) or KCN (1 mM). The concentration of ATP was determined by a luminometer (Orion II; Berthold Detection Systems, Pforzheim, Germany) using the ATP Bioluminescence assay kit CLS II (Roche Applied Science, Basel, Switzerland).

## Cell morphology and immunofluorescence assay

For the immunofluorescence assay, the cells were first treated with 5 mM bathophenanthroline disulphonic acid (BPS), a metalloprotease inhibitor, to stabilize the surface proteins 24 hours prior harvesting [28]. Then, cells were harvested and fixed in 3.7% formaldehyde/PBS for 10 minutes at room temperature. The cell suspension was applied to the polylysine-coated coverslip (Sigma-Aldrich, St. Louis, MO). Then, the coverslips were incubated with primary pAb anti-procyclin (1:400) and anti-BARP (1:400) followed by incubation with Alexa Fluor 488–conjugated goat anti-rabbit secondary antibody. To detect metacyclic cells, the RBP6-induced cells ($5 \times 10^6$) were resuspended in 80 μL media and supplemented with 20 μL of Dextran, Alexa Fluor 568; 10,000 Mw (Thermo Fisher Scientific Waltham, MA). After 1 hour, the cells were fixed by formaldehyde, applied to the polylysine-coated coverslip. The coverslips were then mounted on a glass slide with ProLong Gold Antifade Mountant (Thermo Fisher Scientific Waltham, MA). Images were taken with the fluorescent microscope (Axioplan 2 imaging Universal microscope, Zeiss, Oberkochen, Germany) with a CCD camera (Olympus DP73). To determine cells by their morphology, 100 cells were counted and assigned to a certain cell type based on their size, shape, position of the kinetoplast relative to the nucleus, and position of the kinetoplast relative to the posterior end of the cell (S1 Data).

## Flow cytometry analysis

The $2 \times 10^7$ cells were treated with 5 mM BPS, harvested (1,500*g*, 10 minutes, RT), and resuspended in 200 μL of 1× PBS pH 7.4. The cells were fixed by the addition of 7.4% formaldehyde

in 1× PBS pH 7.4 for 15 minutes at RT. Subsequently, the cells were washed 3 times in 1× PBS, labeled with polyclonal rabbit anti-BARP (1:400) and anti-procyclin (1:400) in 1% BSA for 1 hour at RT, followed by staining with Alexa Fluor 488–conjugated goat α-rabbit (1:400) antibody in 1% BSA. Samples were then washed, resuspended in 1 mL of 1× PBS pH 7.4 and analyzed by flow cytometry. Data were evaluated using BD FACSDiva software.

## LC-MS metabolomic analysis

For the LC-MS metabolomic analysis, the sample extraction was performed as described previously [81,82]. Briefly, $5 \times 10^7$ cells were used for each sample. Cells were rapidly cooled in a dry ice/ethanol bath to 4°C, centrifuged at 1,300$g$, 4°C for 10 minutes, washed with 1× PBS, and resuspended in extraction solvent (chloroform:methanol:water, 1:3:1 volume ratio). Following shaking for 1 hour at 4°C, samples were centrifuged at 16,000$g$ at 4°C for 10 minutes, and the supernatant was collected and stored at −80°C. The analysis was performed using separation on 150 × 4.6 mm ZIC-pHILIC (Merck, Kenilworth, NJ) on Dionex UltiMate 3000 RSLC (Thermo Fisher Scientific Waltham, MA) followed by mass detection on an Orbitrap Fusion mass spectrometer (Thermo Fisher Scientific Waltham, MA) at Glasgow Polyomics.

## Statistical analysis

The number of replicates, controls, and statistical tests are in accordance with published studies employing comparable techniques and are generally accepted in the field. Statistical differences were analyzed with Prism software (version 8.2.1, GraphPad software). Comparisons of two groups were calculated with two-tailed paired *t* test. A *P* value of less than 0.05 was considered statistically significant. Quantitative mass spectrometry experiments were performed in four biological replicates.

## Supporting information

**S1 Fig. PCA shows high reproducibility of replicates and consecutive progression of RBP6$^{OE}$-induced differentiation that can be described by the first two components, PC1 and PC2.** PCA, principal component analysis; RBP6, RNA binding protein 6.
(TIF)

**S2 Fig. RBP6$^{OE}$ transcriptomes highly correlate with the published time course of RBP6 induction [31].** The genes with fold change larger than 2 or smaller than 0.5 (with Benjamini-Hochberg–corrected *P* values smaller than 0.05) are highlighted in red. RBP6, RNA binding protein 6.
(PDF)

**S3 Fig. RBP6$^{OE}$ transcriptomes become increasingly more similar to pure metacyclics [26].** The Pearson correlation values reflect the similarity of the transcriptomes of the respective time points to the two trypomastigote types—procyclic and metacyclic forms. RBP6, RNA binding protein 6.
(PNG)

**S4 Fig. Differentiation proteomics.** The heatmap encompassing 5,227 z-scored LFQ quantified protein groups illustrates significant proteome remodeling during RBP6-induced differentiation. LFQ, label-free quantification; RBP6, RNA binding protein 6.
(PDF)

**S5 Fig. Differentiation proteomics.** PCA of the proteomic samples shows the reproducibility of replicates. PCA, principal component analysis.
(PDF)

**S6 Fig. Heatmap showing log2 fold change of average LFQ intensities of all complex I, III, IV, and V subunits identified in RBP6-induced samples compared to uninduced (day 0).** The color key differs for each map and is always located below the heatmap. LFQ, label-free quantification; RBP6, RNA binding protein 6.
(JPG)

**S7 Fig. Oxygen consumption rates in live RBP6$^{OE}$ cells in the absence of substrate.** The black lines show a decreasing concentration of oxygen in the buffer (left y-axis), while the red line shows O$_2$ flux per cell (right y-axis). Inhibition of AOX-mediated respiration was induced by addition of SHAM. The addition of KCN inhibited respiration via complex IV. AOX, alternative oxidase; KCN, potassium cyanide; RBP6, RNA binding protein 6; SHAM, salicylhydroxamic acid.
(PDF)

**S8 Fig. Heatmap showing log2 fold change of average LFQ intensities of selected proteins involved in redox metabolism and mitochondrial carrier proteins identified in RBP6-induced samples compared to uninduced (day 0).** The color key differs for each map and is located below the heatmap. LFQ, label-free quantification; RBP6, RNA binding protein 6.
(PDF)

**S1 Table. RNA-Seq results for RBP6$^{OE}$ cells undergoing differentiation.** Sheet 1 contains gene IDs for *T. brucei* strain 427 (https://tritrypdb.org/tritrypdb/), their respective best orthologs from *T. brucei* strain 927, and RPKM values for each sample. The experiment was performed in quadruplicates for time points 0, 2, 3, 4, 6, and 8 days upon RBP6 induction. Analyses using R version 3.4.3 and DESeq2 version 1.18.1 were used to identify differentially expressed mRNAs, which were identified using a threshold of Benjamini-Hochberg–corrected *P* values <0.05. RBP6, RNA binding protein 6; RPKM, reads per kilobase of transcript, per million mapped reads.
(XLSX)

**S2 Table. Cluster assignment—transcriptomics.** Gene IDs belonging to four different clusters from time-course expression profiling based on K-medoids. GO enrichment analyses performed using GO Term annotations TriTrypDB-36_TbruceiLister427_GO.gaf from TriTrypDB version 36 and Fisher's exact test. GO, Gene Ontology.
(XLSX)

**S3 Table. Comparison of RNA-Seq data of RBP6$^{OE}$ cells (time points 0, 2, 3, 4, and 6 days) with the time course of RBP6 induction published in [31].** Sheets contains gene IDs for *T. brucei* strain 427 (https://tritrypdb.org/tritrypdb/), their respective best orthologs from *T. brucei* strain 927, log2 fold change, Benjamini-Hochberg–corrected *P* values, and RPKM values for each sample. RBP6, RNA binding protein 6; RPKM, reads per kilobase of transcript, per million mapped reads.
(XLSX)

**S4 Table. Proteomic analysis of RBP6$^{OE}$ cells undergoing differentiation.** Sheet 1 contains Tb427 and Tb927 gene IDs and descriptions for 5,227 protein groups identified by a minimum of 2 peptides (1 unique) and present in at least two out of four replicates. Other sheets contain protein groups differentially expressed (log2 fold change <−1, log2 fold change >1). RBP6,

RNA binding protein 6.
(XLSX)

**S5 Table. Cluster assignment—proteomics.** Gene IDs belonging to six different clusters from time-course expression profiling based on K-medoids. GO enrichment analyses performed using GO Term annotations TriTrypDB-36_TbruceiLister427_GO.gaf from TriTrypDB version 36 and Fisher's exact test. GO, Gene Ontology.
(XLSX)

**S6 Table. Metabolomic analysis of RBP6$^{OE}$ cells undergoing differentiation.** LC-MS metabolomic data. LC-MS, liquid chromatography–mass spectrometry; RBP6, RNA binding protein 6.
(XLSX)

**S1 Video. In vivo measurements of the catalase activity.** The activity of the catalase was detected using a simple visual activity test. A total of $5 \times 10^7$ parasites were resuspended in 100 μL of PBS and placed on a microscopic slide. A total of 20 μL of 3% $H_2O_2$ was added to the cells, mixed, and the formation of oxygen (bubbles formation) was monitored visually. PBS, phosphate-buffered saline.
(MP4)

**S1 Data. All experimental data used to generate graphs of this manuscript.**
(XLSX)

**S1 Raw Images. Original images supporting blot results reported in Figs 1, 3, 6, 7 and 10.**
(PDF)

## Acknowledgments

We thank Martina Slapničková, Jasmin Cartano, and Anja Freiwald for technical assistance.

## Author Contributions

**Conceptualization:** Eva Doleželová, Brian Panicucci, Christian J. Janzen, Alena Zíková.

**Data curation:** Eva Doleželová, Michaela Kunzová, Mario Dejung, Michal Levin, Brian Panicucci, Clément Regnault, Christian J. Janzen, Michael P. Barrett, Falk Butter, Alena Zíková.

**Formal analysis:** Mario Dejung, Michal Levin, Clément Regnault, Christian J. Janzen, Michael P. Barrett, Falk Butter, Alena Zíková.

**Funding acquisition:** Michael P. Barrett, Falk Butter, Alena Zíková.

**Investigation:** Eva Doleželová, Michaela Kunzová, Brian Panicucci.

**Resources:** Michael P. Barrett, Falk Butter, Alena Zíková.

**Supervision:** Alena Zíková.

**Validation:** Eva Doleželová, Alena Zíková.

**Visualization:** Alena Zíková.

**Writing – original draft:** Mario Dejung, Michal Levin, Michael P. Barrett, Falk Butter, Alena Zíková.

**Writing – review & editing:** Mario Dejung, Michael P. Barrett, Falk Butter, Alena Zíková.

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
