## [Editor Report · Decision Letter 0]

6 Dec 2019

Dear Dr Zíková, 

Thank you for submitting your manuscript entitled "Cell-based and multi-omics profiling reveals dynamic metabolic repurposing of mitochondria to drive developmental progression of Trypanosoma brucei" for consideration as a Research Article by PLOS Biology.

Your manuscript has now been evaluated by the PLOS Biology editorial staff as well as by an academic editor with relevant expertise and I am writing to let you know that we would like to send your submission out for external peer review.

Please re-submit your manuscript within two working days, i.e. by Dec 08 2019 11:59PM.

Kind regards,

Lauren A Richardson, Ph.D

Senior Editor

PLOS Biology

---

## [Decision Letter · Decision Letter 1]

10 Jan 2020

Dear Dr Zíková,

Thank you very much for submitting your manuscript "Cell-based and multi-omics profiling reveals dynamic metabolic repurposing of mitochondria to drive developmental progression of Trypanosoma brucei" for consideration as a Research Article at PLOS Biology. 

As you will read, the reviewers appreciated many aspects of your work. However, they do raise concerns that will need to be addressed in a revision. Of particular note, the reviewers note missing controls and discrepancies in the data that need to be investigated further. After discussion with the Academic Editor, we have decided that your manuscript would be better suited for further consideration as a 'Methods and Resources' article. During your revision, please highlight the value of these datasets for further studies and make them as accessible as possible. Lastly, the Academic Editor notes that in lines 183-187 it is stated that “cluster 1 includes proteins that are being upregulated…in agreement with the transcriptomic data, it contains enzymes involved in energy metabolism….” but that this is not reflected in the figure.

In light of the reviews (below), we will not be able to accept the current version of the manuscript, but we would welcome re-submission of a much-revised version that takes into account the reviewers' comments. We cannot make any decision about publication until we have seen the revised manuscript and your response to the reviewers' comments. Your revised manuscript is also likely to be sent for further evaluation by the reviewers.

We expect to receive your revised manuscript within 2 months. 

**IMPORTANT - SUBMITTING YOUR REVISION**

*NOTE: In your point by point response to to the reviewers, please provide the full context of each review. Do not selectively quote paragraphs or sentences to reply to. The entire set of reviewer comments should be present in full and each specific point should be responded to individually, point by point.

*Re-submission Checklist*

*Published Peer Review*

*PLOS Data Policy*

*Blot and Gel Data Policy*

Sincerely,

Lauren A Richardson, Ph.D

Senior Editor

PLOS Biology

REVIEWS:

Reviewer #1: 

Following uptake into the tsetse fly, African trypanosomes multiply as procyclic cells. After differentiation into epimastigote forms, infectious trypomastigotes are formed which are again transmitted to the mammalian host, closing the digenetic life cycle. The work submitted by Dolezelova et al. takes advantage of a previous finding that overexpression of RBP6, a specific RNA-binding protein, induces the differentiation process in vitro. The authors present thorough transcriptome, proteome and metabolome analyses with the aim to get a deeper insight in the changes occurring during the differentiation processes in the insect vector of Trypanosoma brucei. Most remarkably, they show that ectopic expression of catalase in the cytosol of procyclic parasites halts the in vitro-induced differentiation and suggest that hydrogen peroxide may act as signaling molecule.

The data presented are new and of significant interest especially for scientists working on the differentiation of African trypanosomes and related parasites. However, the manuscript requires a number of corrections and alterations. 

Major points:

How long where the cells grown in the absence of glucose and presence of N-acetyl glucosamine before RBP6 overexpression was induced? Can the authors exclude that non-differentiating procyclic cells show similar alterations (protein expression profile, progressive proline and succinate consumption, ATP production etc.) when they are long-term cultured under these conditions? Non-induced cells cultured for six or eight days in the absence of glucose but presence of N-acetyl glucosamine should have been included for control.

Figure 5, why is the O2-flux at day 0 in (A) resting cells as high as in (B) glycerol-3-phosphate-induced cells? 

Figure 6 and text, lines 265-269, please clarify the section and remove redundancies. The Western blots of complex IV at day 4 do not show any significant changes; lines 270-275, the description refers exclusively to day 2. Please comment why at day 6, the activity of complex II is again as low as on day 0.

Figure 8F, please explain the extremely high ADP/ATP ratio of 2, even on day 0. Usually, non-stressed cells have an ADP/ATP ratio of 0.1 to 0.2. 

Figure 10, C, please include the number of experiments and describe the Western blot in the legend, including the time point at which the samples were taken; D, please include the number of experiments; E and text, line 387, the statement about the relative AOX expression is not convincing as in the RBP6-OE-catalase sample (left blot), the AOX level in bloodstream cells is higher than in the RBP6-OE sample (right blot); and line 389, why are the aconitase levels in the catalase-expressing cells lower than in the cells not expressing catalase?; lines 392-397, the section is very speculative, especially " including signaling molecules" and should be modified; F, also in RBP6-OE cells without catalase (Fig 7E), the TMRE fluorescence increases upon induction; G and B, please use the same scale for the y-axis. 

Lines 45-46, should it read "aerobic cells"? Otherwise, the sentence is not clear.

Line 225, the statement that glucose-6-phosphate isomerase functions only in the gluconeogenesis direction is not correct. Please define SBPase.

Lines 296-298, the sentence starting with "Accordingly" is not clear. 

Minor points:

For readers not specialized in the omics-field, the methodology should be described in more detail and the abbreviations (e.g. PCA; K-means, LFQ) defined in the text and/or figure legends. The same abbreviation should be used throughout the manuscript (e.g. BF or BSF, PF or PCF; non-induced or uninduced; in Figure 3, glucose-6-phosphate isomerase is given as "P-6-glu isom" and in Figure 4 as Glu-6-P-ISO. In general, "glucose" should be abbreviated as "glc" not "glu", the three-letter code for glutamate). "Oxal" should be replaced by "oxaloacetate" etc.

Please replace in line 36, "acted" by "appears to act" and in line 37, "exogenous" by "ectopic".

Line 183, please define the clusters 2, 5 and 6.

Line 239, please replace "synthase" by "synthetase".

Line 313, please include (Fig 8A).

Line 315, AOX is not described in Fig 6; should it be Fig3B?

Lines 349-351, please describe more precisely which changes are consistent with the scheme in Fig 4?

Line 355, the authors describe that the reduced form of trypanothione was not detected, what is with the oxidized form?

Line 416, please modify the sentence. It should read "we detected lower levels of proline and glutamate which may suggest a higher consumption".

Line 434, replace "synthetase" by "synthase".

Line 482-484, to get a deeper insight in the subcellular origin of hydrogen peroxide, catalase could be expressed in the mitochondrial matrix.

Figure 2D, it should read "Protein phosphorylation"

Figure 3A, "day" and "log 2-fold", respectively, should be included in front of the numbers. 

Figure 4, the use of different colors for the upregulated reactions would facilitate reading. All abbreviations should be defined in the legend. Pyruvate is given twice in grey boxes, ones without, please clarify.

Figure 5C, what is the meaning of s.d. for only two values?

Figure 7 legend, include after "cells" "that", see also comment for Figure 5

The supplementary figures S1-S5 were missing in the manuscript provided and thus were not evaluated.

---------------

Reviewer #2: 

Trypanosoma brucei lives in two hosts, mammals and tsetse flies, and adapts its metabolism to its different environments. Until recently, the majority of life-cycle stages from tsetse could not be cultured. In this study the authors exploit a system first described by Kolev et al., over-expression of the RNA-binding protein RBP6. This drives differentiation in culture from the procyclic (fly midgut) form, via a series of intermediate life-cycle stages, to the metacyclic form which is infectious for mammals. In contrast to Kolev et al., the authors use a glucose-free medium, which appears to make differentiation more efficient. This study provides time courses of the transcriptome, proteome and metabolome, with a focus on the mitochondrion which changes during development. Interestingly, reactive oxygen species seem to play a role in differentiation as expression of catalase (an enzyme not normally present in trypanosomes) blocks life-cycle progression in cultures. 

I appreciate this work as it contains a wealth of data, but I would have liked to have seen it analyzed more extensively. The authors focus on the metabolic capability and remodeling of the mitochondrion, an organelle that is present in almost all eukaryotes. Nevertheless, for the following reasons, I am in two minds about whether it fits the scope of PLoS Biology or if the data would be more appropriate for another PLoS journal (e.g. PLoS Pathogens):

1. The work sheds most light on differentiation and could identify new stages or intermediate stages in the life cycle. It might then be easier to attribute metabolic pathways to specific stages of the life cycle. 

2. There are nuances in the system that are difficult for non-parasitologists to appreciate. The differentiating cultures consist of mixtures of cells, some of which do not usually coexist in one tissue. The data thus reflect these mixtures overlaid upon each other. This differs from a previous study by Cristiano et al., in which in vitro-derived metacyclics were purified prior to transcriptomic and proteomic analyses. Incidentally, their data strongly suggested that metacyclic forms and bloodstream forms are metabolically similar. Importantly, by day 8 in the current study, there is a considerable proportion of "procyclic cells" (more about this below). This means that the cells in culture are either dedifferentiating and/or that there remaining epimastigotes are non-dividing and are being overgrown by procyclics. This may also explain why the clusters of co-expressed transcripts almost all change their trend on day 8.

3. Although the complex life cycle is clearly described, it is not sufficiently clear how different stages were categorized. For example, in Figure 1A, panel day 2 epimastigotes: only one cell has the classic anteronuclear kinetoplast. One cell has a procyclic configuration. The other two might be transitioning as the kinetoplast is very close to or over the nucleus. 

Intriguingly, if the majority of cells on day 2 are epimastigotes, they are not expressing BARP proteins (see Figure 10E). While it is known what salivary gland epimastigotes express BARPs, it is not known if they are expressed by epimastigotes in the foregut. Might there be two populations of epimastigotes in these cultures and are their mitochondria and metabolism different? The authors may be able to extract some more information from their data sets.

4. The procyclic forms on day 8 do not reflect the natural life cycle. They don't seem to express GPEET procyclin protein or mRNA. What about the other procyclin marker, EP? This information could be extracted from the RNA-seq data. If they are procyclic forms, it should be made very clear that this is a culture artefact. Alternatively, it might be better to remove most of the day 8 data. 

5. RNA-seq data (Table S1). The gene IDs are from the 2018 version of T. brucei 427. Although this makes sense, because this cell line is a derivative of 427, it makes it harder to browse the data when it comes to differentially expressed genes. Would it be possible to include the orthologs in T. brucei 927 and gene names in all of the tables?

6. The role of ROS and the effect of ectopic catalase are intriguing. It was shown recently that trypanosomes expressing catalase were less successful at establishing midgut infections. Based on the data shown here (presence of GPEET absence of BARP), it looks like the block in differentiation occurs much earlier than the transition from epimastigotes to metacyclic forms. Analyzing the cells in these cultures by immunofluorescence or flow cytometry, in addition to morphology, would be informative. Are the majority procyclic forms (expressing GPEET) or is this a sub-population?

7. The metabolome is hardly discussed apart from metabolites that change in concentration over the course of the experiment. Can more information be extracted? Are there indications of unusual pathways or missing pathways?

Minor points:

8. "Morphotype" is not a word that is normally applied to trypanosomes. Should this be life-cycle stage? Monoformic (monomorphic?). This refers to bloodstream forms that have been syringe-passaged until slender forms lose the capacity to differentiate to stumpy forms. 

9. The word "and" is missing from line 119. It should be 

 … typically smaller than epimastigotes and PCF, and are ….

---------------

Reviewer #3: 

In this manuscript Doleželová et al took advantage of a previously established in vitro differentiation system based on the over-expression of the RNA binding protein 6 (RBP6) to follow changes at the RNA, protein and metabolite levels during the development of Trypanosoma brucei procyclics to infectious metacyclics. Although this development takes place in the tseste fly vector, investigations in the fly are not feasible due to difficulties in acquiring enough parasites, especially for proteomic and metabolomic studies. Thus, the RBP6 over-expression system provides an ideal model system. The authors performed a careful multi-omics study and provide convincing evidence for redirection of electron flow from the cytochrome mediated pathway to an alternative oxidase. This is a rich set of data that will be very valuable to the community. Most importantly, they not only provide insights into the mechanisms of the parasite´s mitochondrial rewiring, but also strengthen the emerging concept that mitochondria act as signaling organelles through the release of reactive oxygen species to drive cellular differentiation.

There are only a few minor points that will need attention.

1. A recent publication (MBP 224, 50-56, 2018) describes an RNA-seq analysis of the time course of RBP6 induction. Although the data presented here are more detailed due to the higher efficiency of differentiation, a comparison of the results is warranted.

2. The sentence starting on line 333 needs a reference. 

3. Line 496: "Even though RBP6OE is not the physiological route to differentiation of the midgut PCF trypomastigote, " What do the authors mean? Do they have evidence for the physiological route to differentiation?

---

## [Decision Letter · Decision Letter 2]

22 Apr 2020

Dear Dr Zíková,

Thank you for submitting your revised Methods and Resources article entitled "Cell-based and multi-omics profiling reveals dynamic metabolic repurposing of mitochondria to drive developmental progression of Trypanosoma brucei" for publication in PLOS Biology. I have now obtained advice from one of the original reviewers and have discussed these comments with the Academic Editor. 

Based on this review (attached below), we will probably accept this manuscript for publication, assuming that you will modify the manuscript to address the remaining points raised by the reviewers. You should discuss the remaining points and, if required, modify the manuscript. This includes the ADP/ATP issue, which remains unresolved. If you cannot explain these levels, you should state this clearly in the manuscript.

Please also make sure to address the data and other policy-related requests noted at the end of this email.

We expect to receive your revised manuscript within two weeks. Your revisions should address the specific points made by each reviewer. In addition to the remaining revisions and before we will be able to formally accept your manuscript and consider it "in press", we also need to ensure that your article conforms to our guidelines. A member of our team will be in touch shortly with a set of requests. As we can't proceed until these requirements are met, your swift response will help prevent delays to publication.

*Copyediting*

*Published Peer Review History*

*Early Version*

*Submitting Your Revision*

Sincerely,

Ines

--

Ines Alvarez-Garcia, PhD

Senior Editor

PLOS Biology

Carlyle House, Carlyle Road

Cambridge, CB4 3DN

+44 1223–442810

DATA POLICY:

Fig. 1B, E; Fig. 2B, C, D, G; Fig. 5A, B, C; Fig. 7A, B, D, E; Fig. 8A, B, C, D, E, F; Fig. 10A, B, C, D, F, G, H, I and Fig. S2A, B

Reviewers' comments

Rev. 2:

I have been asked to address the points raised by reviewer 1 (who is currently unavailable) as well as my own (reviewer 3).

Points raised by reviewer 1:

1. Most of the major points have been dealt with, including the issue of pre-adapting the cells to glucose-free medium. I am not sure if the reviewer would have been satisfied by the responses to his/her points 2 and 4, where the data are now depicted as ratios. These do not really address the questions about absolute values. For point 4, the authors used two independent kits to determine ADP/ATP levels, which adds credence to their findings, but they avoid mentioning that the levels of ADP are unusually high by using a "normalised" ratio of 1.

Points that I raised.

2.Again, most have been dealt with satisfactorily. There is one discrepancy, however. In the Western blot in Figure 10E, BARP is not detectable in cells expressing catalase, but is detected in Figure 10F (flow cytometery) in about 10% of cells. This may be a sensitivity issue, but it should be addressed.

3. Please check the revised version for typos, etc. There are several places where spaces are missing between words, letters were transposed or singular and plural didn't match.

Media should be medium (if it is only one). "Mixture" might be more appropriate than "blend" to describe the cultures.

---

## [Editor Report · Decision Letter 3]

27 May 2020

Dear Dr Zíková,

On behalf of my colleagues and the Academic Editor, André Schneider, I am pleased to inform you that we will be delighted to publish your Methods and Resources in PLOS Biology. 

Early Version

PRESS 

Kind regards,

Vita Usova 

Publication Assistant, 

PLOS Biology

on behalf of

Ines Alvarez-Garcia,

Senior Editor

PLOS Biology